# Computer-Aided Design of Traditional Jigs and Fixtures

**Abdullah D. Ibrahim** [1] , **Hussein M. A. Hussein** [2,3] , **Ibrahim Ahmed** [4] , **Emad Abouel Nasr** [5] , **Ali Kamrani** [6] and **Sabreen A. Abdelwahab** [7,*]

1   Senior Lab Engineer, Process and Product Development Unit, American University in Cairo, Cairo 11835, Egypt; Abdullah.Hassanin@aucegypt.edu
2   Mechanical Engineering Department, Faculty of Engineering, Helwan University, Cairo 11732, Egypt; hussein@h-eng.helwan.edu.eg
3   Mechanical Engineering Department, Faculty of Engineering, Ahram Canadian University, Giza 12451, Egypt
4   Automotive Technology Department, Faculty of Technology and Education, Helwan University, Cairo 11732, Egypt; ilmahmed1968@yahoo.co.uk
5   Industrial Engineering Department, College of Engineering, King Saud University, Riyadh 11421, Saudi Arabia; eabdelghany@ksu.edu.sa
6   Industrial Engineering Department, College of Engineering, University of Houston, Houston, TX 77204, USA; akamrani@uh.edu
7   Production Technology Department, Faculty of Technology and Education, Helwan University, Cairo 11732, Egypt
*   Correspondence: Engsabreenabdallah@gmail.com

**Abstract:** Conventional design of jigs and fixtures has become unsuitable given the requirements of modern technology and complexity and diversity in the production with the rapid update of products. Computer-aided design (CAD) of jigs and fixtures is an effective solution in this direction. The current paper focuses on a computer-aided design of the traditional jigs and fixtures and developed a system containing tailor-made software, created using the Visual Basic programming language and installed on it the viewer screen to show the part. The developed system has been built by connecting Visual Basic programming language to the SolidWorks software on which the part is drawn and saved as STEP AP-203 file format, and the system reads and extracts the data from the STEP AP-203 file. Heuristic rules of feature recognition are pre-prepared for checking the extracted geometric data and deciding which data shape will represent the machining feature; then, the system provides the optimum design of the traditional jigs and fixtures for a group of hollow cylindrical parts that contain a group of cross-holes on the cylinder body, whether perpendicular or offset from the cylinder's axis, (inclined or inclined offset, or blind or through, by applying pre-prepared heuristic rules for the design of traditional jigs and fixtures.

**Keywords:** traditional fixtures; jig and fixture design; cross-hole; data extraction; feature recognition; STEP AP-203 file

## 1. Introduction

Jigs and fixtures are devices which are widely used in manufacture; they provide guidance for the tool and supporting, locating, and holding for the workpiece during manufacturing process. Moreover, they are used for assembling a big number of parts [1]. Jigs and fixtures help in eliminating the individual marking, adjusting the position, and repeating the checking and getting a better quality of manufacturing process. This increases productivity and reduces the time of the operation. The efficient and reliable design and manufacture of jigs and fixtures became more required with the application of computer-aided manufacture (CAM) technology. The manual designing of the jigs and fixtures is considered time-consuming in manufacturing process, and the designer's experience and skill play an important role in the designing of jigs and fixtures [2].

Jigs and fixtures are helping devices which deal with cutting machines such as milling machines, turning machines, drilling machines, grinding machines, and broaching ma-

chines. Welding, assembly, and inspection fixtures are other branches of fixtures which have no deals with the cutting machine. Jigs are machining aid devices which include means of tool guiding in drilling operations, such as drilling and reaming processes. Fixtures are holding devices which are used for holding the workpiece firmly in easy way, and they might contain the means of the cutting tool setting; fixtures are used for milling, turning, grinding, and other manufacturing operations [3].

Jigs and fixtures design is a complex process that requires a good experience and product understanding of the designer and might take many days or longer [4].

The costs related to traditional jigs and fixtures may account for (10–20%) of the total cost of the manufacturing process. These costs include their manufacture, assembly, operation, and design [5].

Automatic feature recognition (AFR) techniques are still limited to asymmetrical and orthogonal features and cannot deal with non-symmetrical and nonorthogonal features and errors made because of combining of AFR techniques. Researchers in the AFR field are still trying to find a fully automated feature recognition algorithm for such features [6].

Jigs and fixtures are still important devices for both traditional manufacturing and modern flexible manufacturing systems (FMS) because they affect the quality and the productivity of the manufacturing process as well as the product cost. Soo Waihong, in 2003, proved that using the time in improving current products and presenting new products is better than wasting it in the manual design and manufacture of jigs and fixtures [7].

Increasing global competition made all manufacturers in the industrial market make the best effort for improving their competitiveness by developing the quality of their products and reducing the production cost and time; therefore, there is a strong need to upgrade the jigs and fixtures design methods for obtaining more efficient design with a lower cost. Developing a computer-aided fixture design (CAFD) technology as an integration of computer-aided design (CAD) and computer-aided manufacture (CAM) can help in achieving this goal which is able to support and simplify the design process [8].

The difference between the traditional fixtures and modular fixtures is that the modular fixtures are primarily used with C.N.C. machines and they do not need to calculate the clamping force during the machining operation. Furthermore, the positioning of the locators and clamps is still a problem within the modular fixtures' assembly. Traditional fixtures design needs several calculation modules for covering several items, such as feature recognition, the required clamping force, the cam design, the locator position, the guide position, selecting the standard components of the fixture, modeling the whole jigs and fixture, and process planning [3,9,10].

Different approaches have been used in jigs and fixtures design, such as case-based reasoning (CBR), rule-based expert system, genetic algorithm (GA), multi-agent approach, and geometric analysis [11,12].

There are few pieces of research found in the literature that cover the traditional jigs and fixture design technique [13,14].

## 2. Literature Review

The literature review section introduces the work that is more closely related to the approach used in the current work as shown in Table 1:

**Table 1.** Summary of major research work on computer-aided design of jigs and fixtures.

| Ref. No. | Authors | System Details | Remarks |
|---|---|---|---|
| [15] | A.S. Kumar, A.Y.C. Nee, and S. Prombanpong (1992) | Used an integrated expert system shell with computer-aided design (CAD). | Developed a complete system for automated fixture design; this system recognizes manufacturing features in the CAD model, determines set-ups, and generates the fixture configuration based on the 3-2-1 locating principle. |

**Table 1.** *Cont.*

| Ref. No. | Authors | System Details | Remarks |
|----------|---------|----------------|---------|
| [9] | A.Y.C. Nee, K. Whybrew, and A.S. Kumar (1995) | Used heuristic rule-based method to generate a fixturing recommendations list which includes the locating and base elements for the fixture. | Presented an expert system for performing the fixture design task and generated a list of fixturing recommendations including the locating and base elements for the fixture. |
| [16] | R.H. Alarco'n, J.R. Chueco, J.M.P. Garcia, and A.V. Idoipe (2010) | Used a knowledge model for the fixture design. | Presented an automatic fixture design system developed by knowledge-based engineering application. |
| [17] | S. Selvakumar, K.P. Arulshri, K.P. Padmanaban, and K.S.K. Sasikumar (2013) | Used an artificial neural networks (ANN)-based algorithm with the design of experiments (DOE). | Proposed a hybrid system for designing the optimum fixture layout to decrease the maximum the workpiece's elastic deformation caused by the clamping and machining forces during the machining operation. |
| [18] | H. Hashemi, A.M. Shaharoun, and I. Sudin (2014) | Used a case-based reasoning method for improving the fixture design process efficiency. | Proposed a system of fixture design in which an appropriate workpiece was found in the first level of the database by applying design requirements. This allowed the proper conceptual fixture design to be achieved by retrieving a related fixture case from the second level. |
| [19] | W. Fu, I. Matthew, and Campbell (2014) | Used a developed hierarchical search system to identify which operations were suitable in terms of manufacturing cost, time, and fixture quality. | Demonstrated a rule-based algorithm that defines fixture design for a set of operations. |
| [3] | H.M.A Hussein, A. Mahrous, A.F. Barakat, and O.M. Dawood (2016) | Used a package including AutoCAD software and Visual Basic programming language. | Developed a systematic framework for a computer-aided design of traditional jigs and fixtures; this system is limited to cylindrical parts having axisymmetrical features. |
| [20] | V. Ivanov, I. Pavlenko, O. Liaposhchenko, O. Gusak, and V. Pavlenko (2018) | Used a technique integrated into a computer-aided fixture design system based on a process-oriented approach. | Proposed a methodology for computer-aided positioning of functional fixture elements regarding the technological part parameters; this system reduced the fixture design time of the drilling, milling, and boring operations and improved the fixtures' components quality. |

After reviewing the previous studies and research, it becomes clear that there is a need for a developed system to recognize the combined and interacted features such as the cross-hole feature and design the appropriate traditional jigs and fixtures.

## 3. Computer-Aided Design of Traditional Jigs and Fixtures

Figure 1 shows a typical example of jig and fixture [21], which includes the locating units on which the workpiece rests for locating it accurately and constrained it in 3:5 degrees of freedom, the clamping units that hold the workpiece during machining for securing the location of the workpiece and the jig guiding elements which guide the cutting tool for obtaining an accurate machining operation [21,22].

There is a relationship between the traditional method of jigs and fixture design and the machine elements design. These devices are primarily dependent on the standard parts, i.e., dowel pins, bolts, cams, etc. [3].

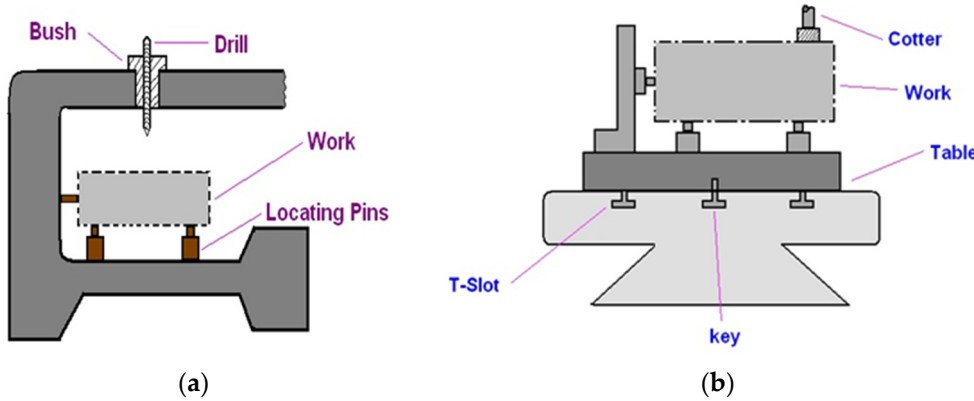

**Figure 1.** Typical example of (**a**) a drilling jig, (**b**) a machining fixture [21].

The jigs and fixtures design process have four phases, which are typically setup planning, jig and fixture planning, unit design, and verification. In the setup planning phase, analyzing the workpiece and machining information is done for determining the suitable position of the locating units and all the setups needed to perform the required machining operations. In the jig and fixture planning phase, six fixturing requirements (physical, tolerance, constraining, affordability, collision prevention, usability) are generated for the setup of a specific manufacturing operation as well as the fixture's layout plan which details the workpiece surface with the positioning of the locating and clamping units. In the unit phase design, the appropriate design generation of units is conducted, such as the design of locating and clamping units for obtaining an accurate location and to overcome the deformation of the fixture components. In the verification phase, the jig and fixture test is carried out. This phase can also be performed before the unit design phase for evaluating the validity of the current fixture system layout [23,24].

CAD part model description in basic geometrical and topological form cannot be used for process planning directly because it requires information extracted from the CAD part model in the form of features, and then, this information is used in process planning [25]. Thus, the problem of the planning process is how to extract, recognize, and manufacture these features.

The extraction and recognition of the manufacturing features in a design database have received significant attention since the mid-1970s; therefore, many standards for data exchange have been developed for solving the design complexity problem and for integrating different CAD systems, such as Initial Graphics Exchange Specification (IGES), Data Exchange Format (DXF), Standard Exchange Transfer (SET) and Standard for Exchange of Product (STEP) format [26,27].

STEP is a standard that gives the obvious and complete representation of the par data during the life cycle, independent of any system [28]. In a STEP file, the specification of product information is standardized by ISO10303, and there are many application protocols (APs) available in it which are used for different applications, such as AP203, AP214, AP224, AP238, etc., as illustrated in Figure 2. The process begins with exporting the model from the STEP AP-203 file. The file is translated for producing an AP-224 data file which is read by MASTERCAM (CNC Software Inc. founded in Massachusetts, USA, 1983). Then, MASTERCAM generates an AP-238 file which contains the geometrical information, features, and tool path for the controller of the machine.

The current work introduces a developed system for designing the traditional jigs and fixtures, in which the workpiece is drawn using SolidWorks software and saved as a STEP AP-203 file format because of its compatibility with most CAD programs. Tailor-made software is used for extracting and recognizing the machining feature of the part and then providing the appropriate jig design of the part.

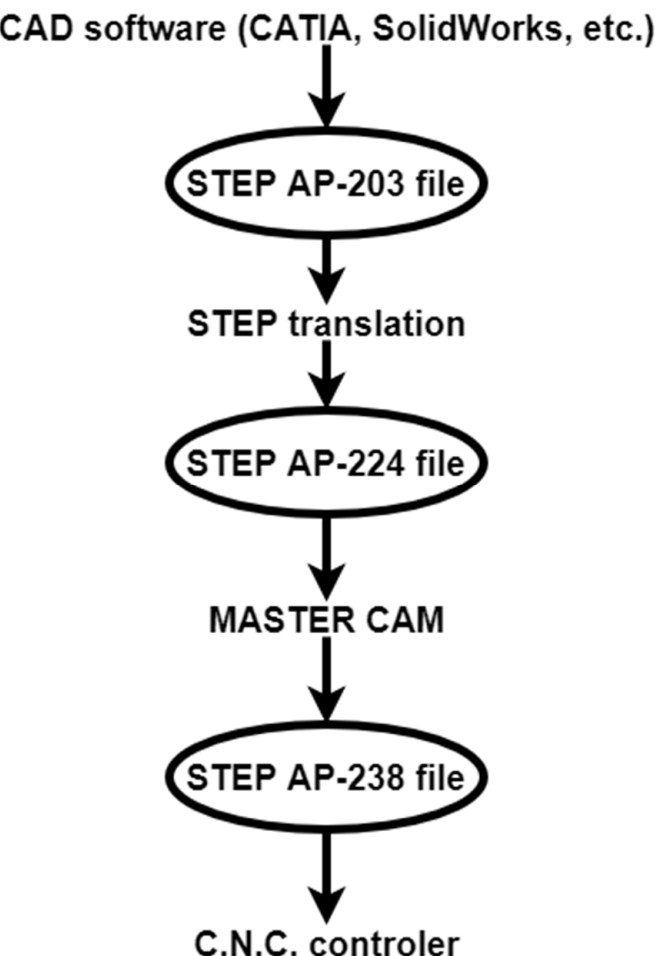

**Figure 2.** Data exchange process, recreated from [27].

## 4. Architecture of the Proposed System

The system proposed in the current work solves three problems: the first one is extracting data from the STEP AP-203 file, the second one is the cross-hole feature recognition in hollow cylinders in different cases, and the third one is providing the suitable jig design of the cylindrical part. The system has been built by connecting the Visual Basic programming language to the SolidWorks software (Dassault Systèmes SOLIDWORKS Corp. Waltham, MA, USA, December 1993). The drawing is sent to Visual Basic after saving it as a STEP AP-203 file. Tailor-made software is installed on Visual Basic which is used for showing the drawing on a special screen. Then, the system extracts the geometrical data and recognizes the cross-hole features of the hollow cylinder [29,30]. Finally, it provides the final design of jigs according to rule-based reasoning. Figure 3 shows the data extraction and feature recognition process [27].

The data extraction and feature recognition processes start with the part that is drawn and saved as STEP AP-203 file format, and the system then reads and extracts the data from the STEP AP-203 file. Heuristic rules of feature recognition are pre-prepared for checking the extracted geometric data and deciding which data shape will represent the machining feature.

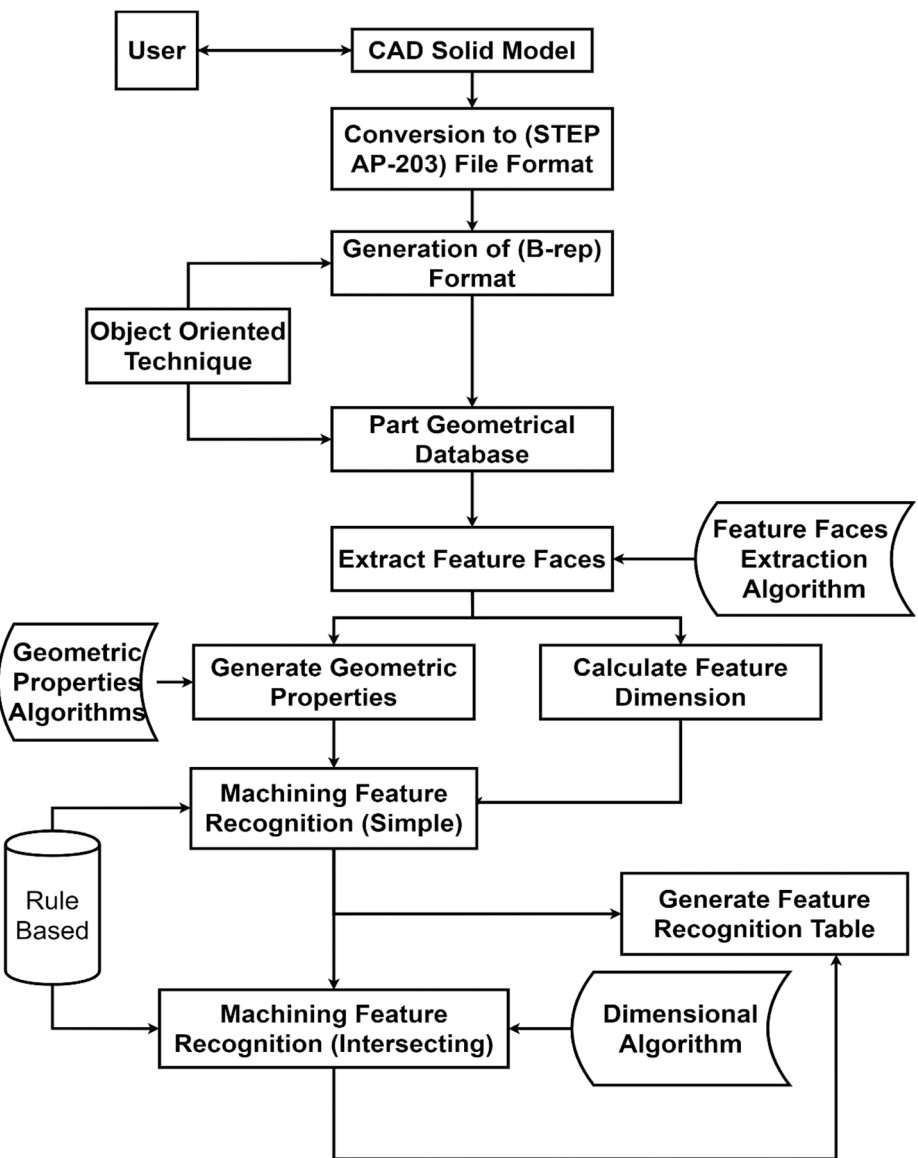

**Figure 3.** Data extraction and feature recognition process [27].

### 4.1. Data Extraction and Feature Recognition

The method used for extracting data from the STEP AP-203 file is the object-oriented approach, in which the system starts to read the STEP AP-203 file and search for (B_Spline_Curve_With_Knots) string which indicates the existence of the cross-hole feature. Then, it traced the data related to that string to identify the orientation of the cross-hole feature. The technique used for representing the part geometrical structure is boundary representation (B-Reb). The approach will specify the type and orientation of all faces using the (B-Rep) database which includes the topological and geometrical information.

Figure 4 shows a flowchart of the feature recognition process, which is based on geometric properties and an object-oriented approach for recognizing cross-hole features.

The flowchart shows that the number of B_Spline_Curve_With_Knots (BSp) indicates that the cross-hole feature is blind or through. For recognizing the offset cross-hole, the system reads the values of vertex (1 and 2) of (BSp), and if both values are ($\neq 0$), this indicates that the cross-hole is offset from the axis of the cylinder. For recognizing the inclined cross-hole, the system reads the directions (Dx, Dy, and Dz) of (BSp); if the values of two directions of these three are (>0), this indicates that the cross-hole is inclined.

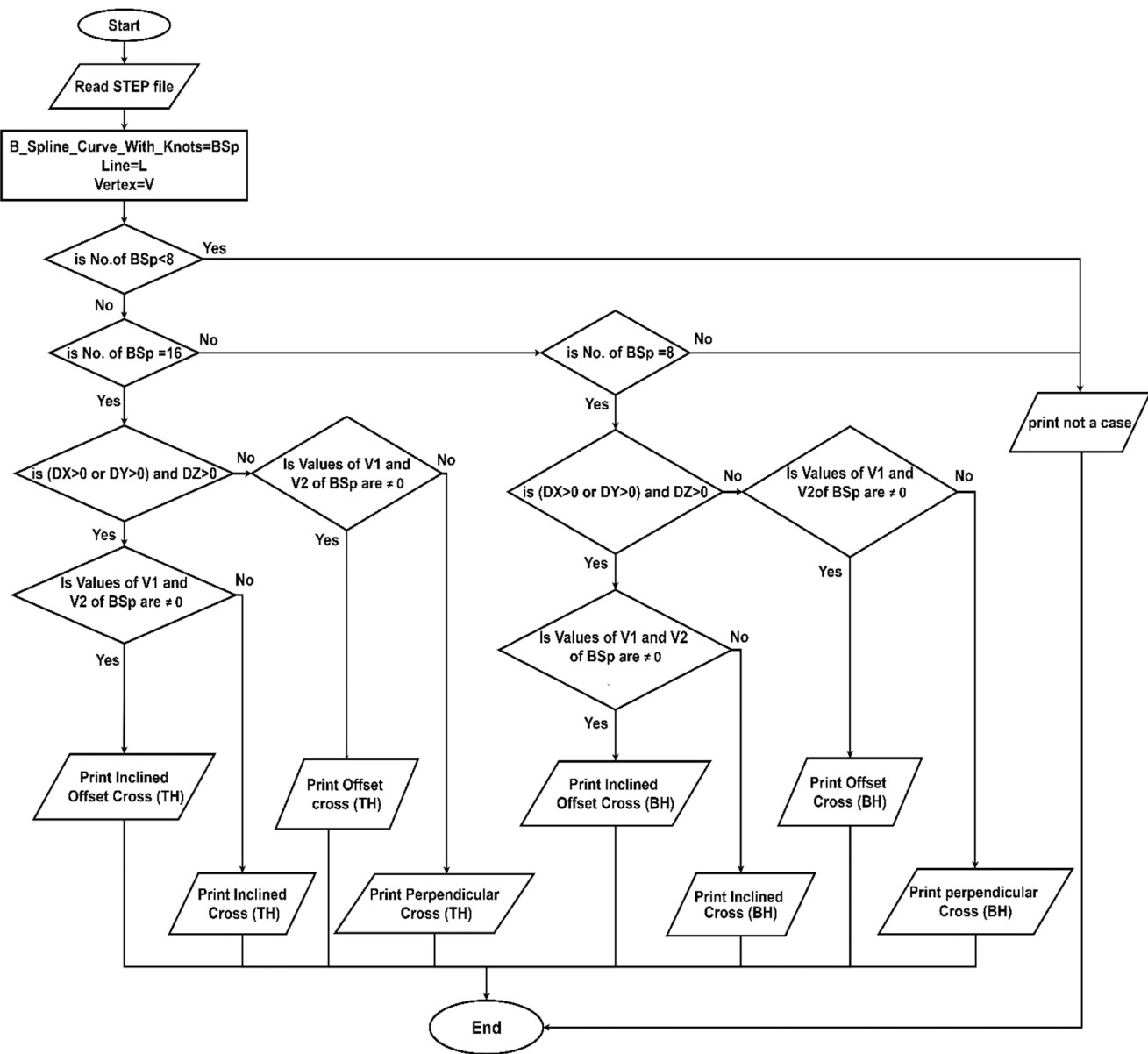

**Figure 4.** Flowchart of automatic feature recognition of cross-holes in hollow cylinders.

### 4.2. Design of Jigs and Fixtures

The jig design process depends on the feature recognition of the part. The proposed system recognizes the cross-hole feature of the cylindrical part which is needed for the design process automatically after checking it, and then the software starts to draw the appropriate jig of the part on the system graphical area according to the geometrical and topological information of the cross-hole feature. Figure 5 shows a flowchart of the traditional jig design.

The flowchart shows that the jig design process depends on the cross-hole feature. Initially, the system read the feature recognition to identify the cross-hole type. If the cross-hole is perpendicular or offset, the system will design a vertical jig according to the design rules. If the cross-hole is inclined or inclined offset, the system will design an inclined jig according to the design rules.

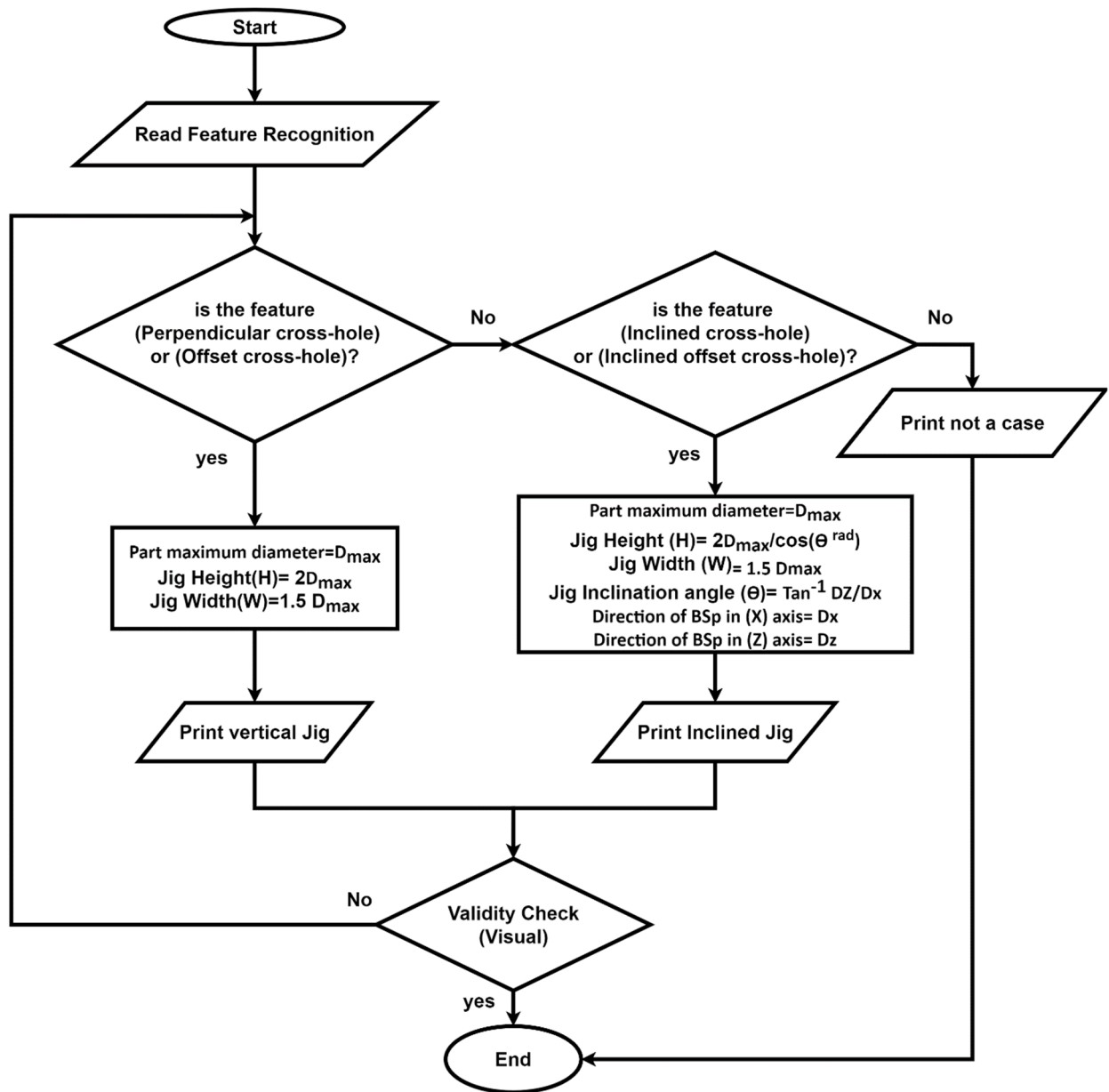

**Figure 5.** Flowchart of computer-aided traditional jigs and fixtures design.

## 5. Validation of the Proposed System

For validating the proposed methodology, it has been tested in both feature recognition and jig design of hollow cylinders with a cross-hole in various orientations. SolidWorks software is used for preparing the part drawing and saving it as a STEP AP-203 file, and then, the file is sent to Visual Basic for showing the part on a special screen. The developed system extracts data from the STEP AP-203 file, recognizes the cross-hole feature, and provides the final design of the jig for the part.

### 5.1. The System Interface

Figure 6 shows the main window of the developed system, including the code files for data extraction, feature recognition, and jigs and fixtures design.

Clicking the (Start) button opens the system interface which contains a group of buttons for importing the STEP AP-203; file, extracting the geometrical data from the STEP AP-203 file, feature recognition, and designing the traditional jigs and fixtures.

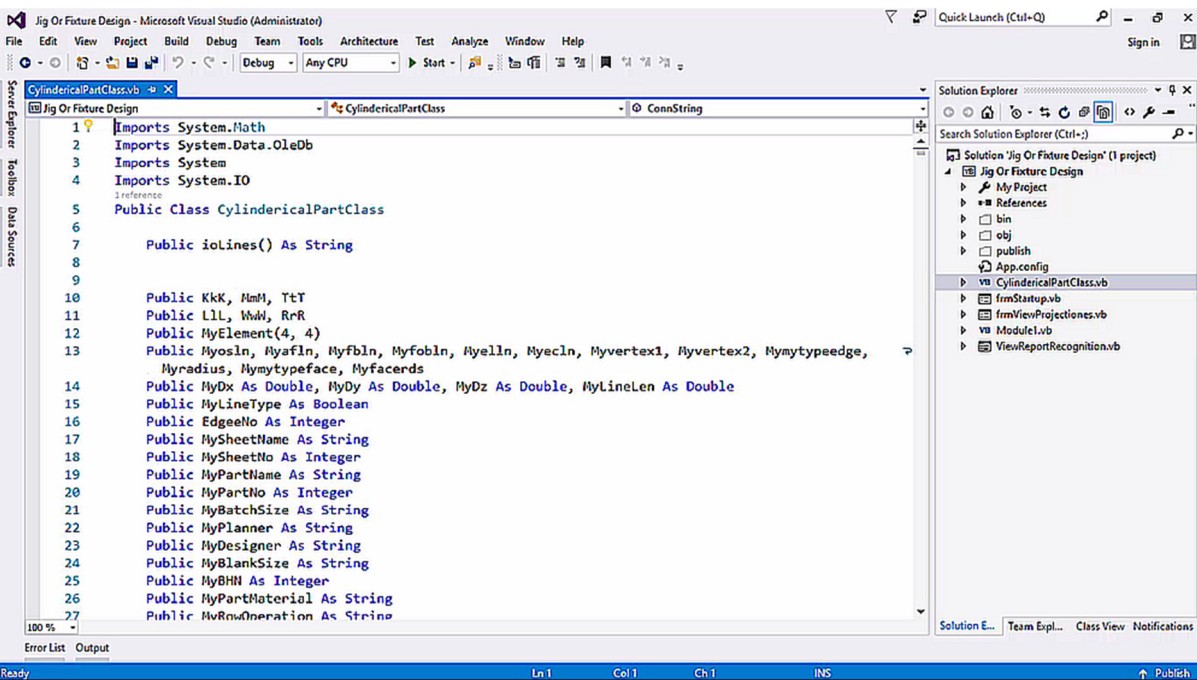

**Figure 6.** The main window of the developed system.

## 5.2. The First Screen

Figure 7 illustrates that the system activates (Import File) and (Exit) buttons and is ready for importing STEP AP-203 files.

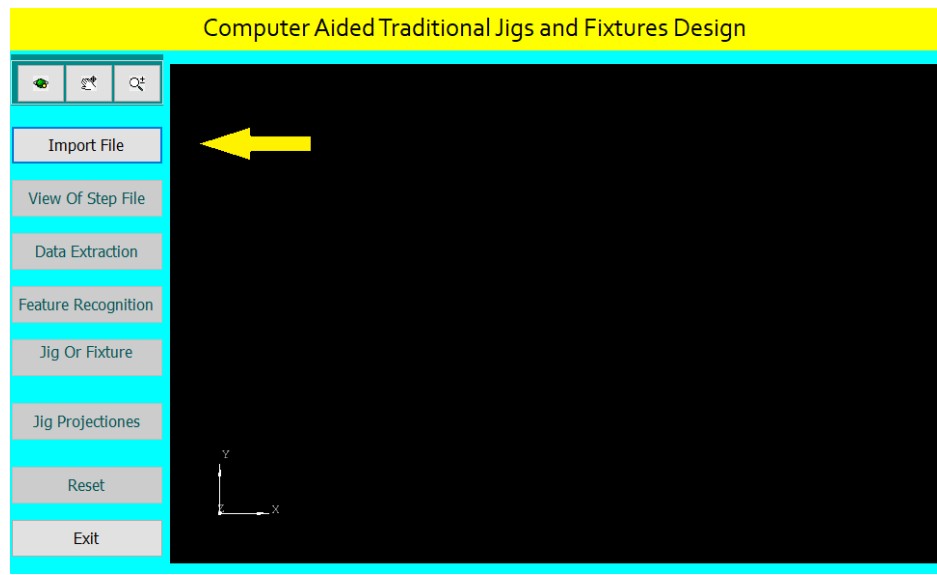

**Figure 7.** The first screen of the system interface.

After clicking (Import File), the system browses the STEP AP-203 file saved on the PC or an external memory for importing it by selecting the file and clicking on the (Open) button as shown in Figure 8.

Once the file is selected and imported, the system activates the other interface buttons and shows the part on its interface screen as illustrated in Figure 9. The system allows zooming in and out, dragging and dropping, and rotating the part.

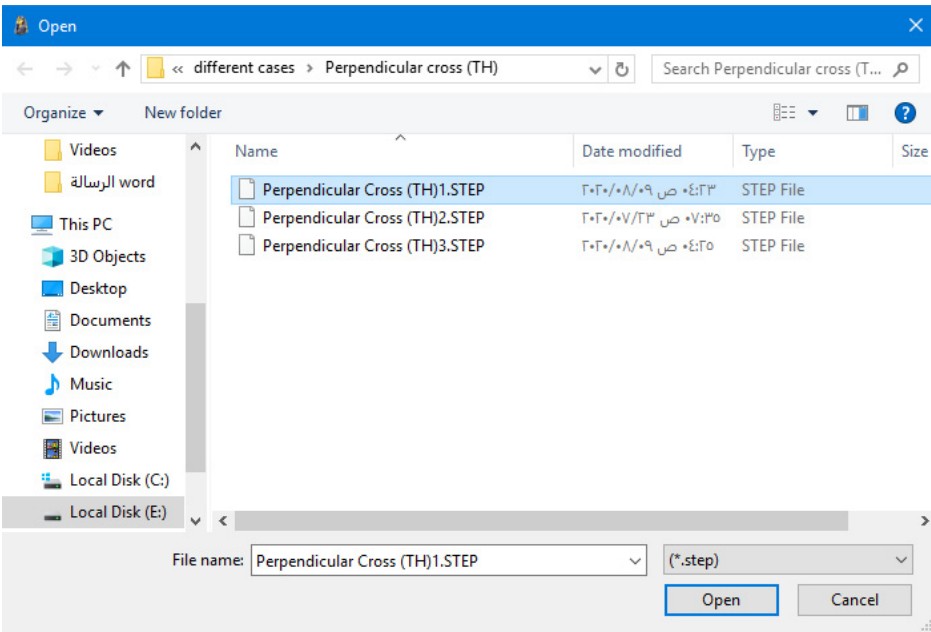

**Figure 8.** Importing STEP AP-203 file.

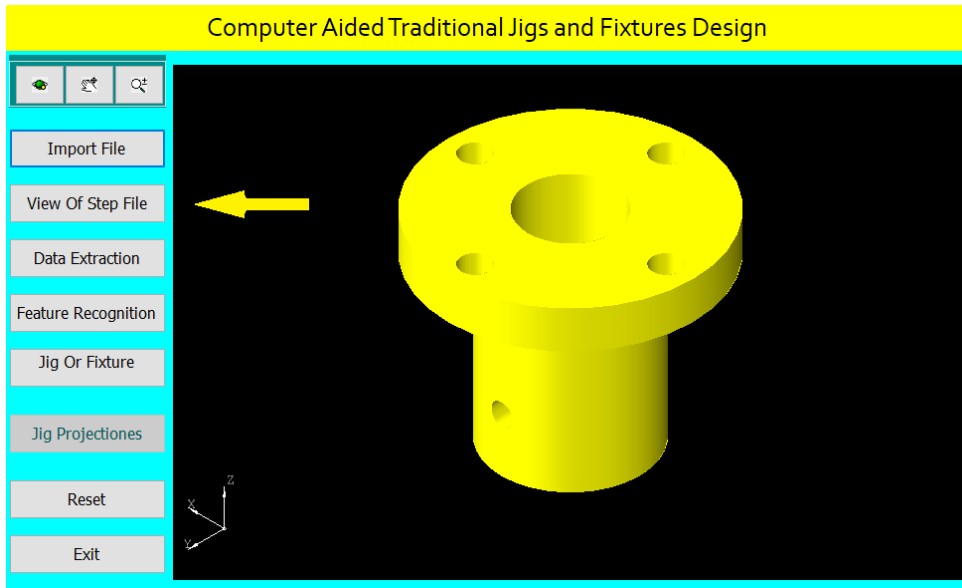

**Figure 9.** The second screen of the system interface.

*5.3. The Second Screen*

After clicking the (View of STEP file) button, the system divides the interface screen into two parts to show both the part and STEP AP-203 file as shown in Figure 10.

*5.4. The Third Screen*

Figure 11 shows the third screen of the system, in which we click the (Data Extraction) button for extracting the geometrical data of the cylindrical part that is needed for feature recognition and traditional jigs and fixtures design.

After clicking on the (Data Extraction) button, the system creates a report of the geometrical data of all the part features according to the rules of data extraction. Figure 12 shows the extracted data report, and this report helps in feature recognition and design of traditional jigs and fixtures for the cylindrical part.

## 5.5. The Fourth Screen

Figure 13 clarifies the fourth screen of the system interface, in which we click the (Feature Recognition) button; then, the system applies the rules to recognize the cross-hole feature and shows a message box of the feature recognitions process.

## 5.6. The Fifth Screen

For designing the appropriate traditional jigs and fixtures for the cylindrical part, we click the (Jig or Fixture) button as shown in Figure 14 in which the system applies the design rules based on the feature recognition process and provides the final design of the traditional jigs and fixtures. The (Jig Projections) button is used to draw the jigs views on the interface screen to show the design details of traditional jigs and fixtures as illustrated in Figure 15.

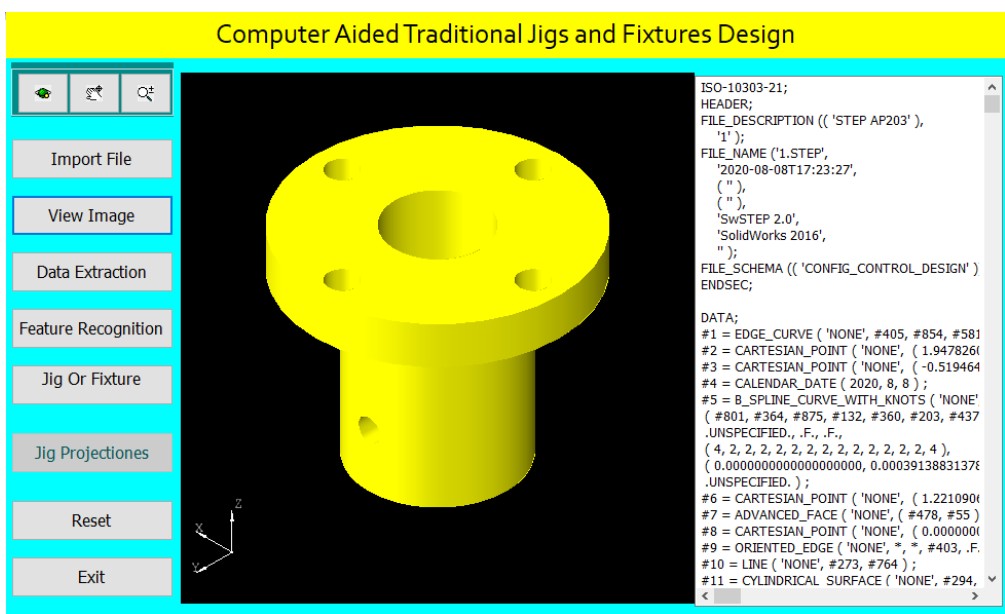

**Figure 10.** The system shows the part and the STEP AP-203 file.

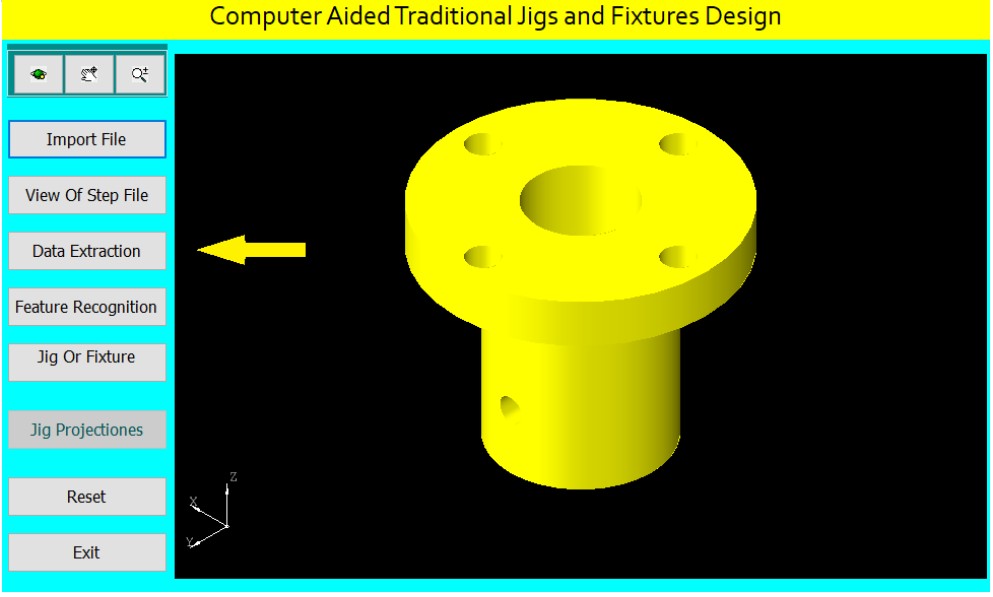

**Figure 11.** The third screen of the system interface.

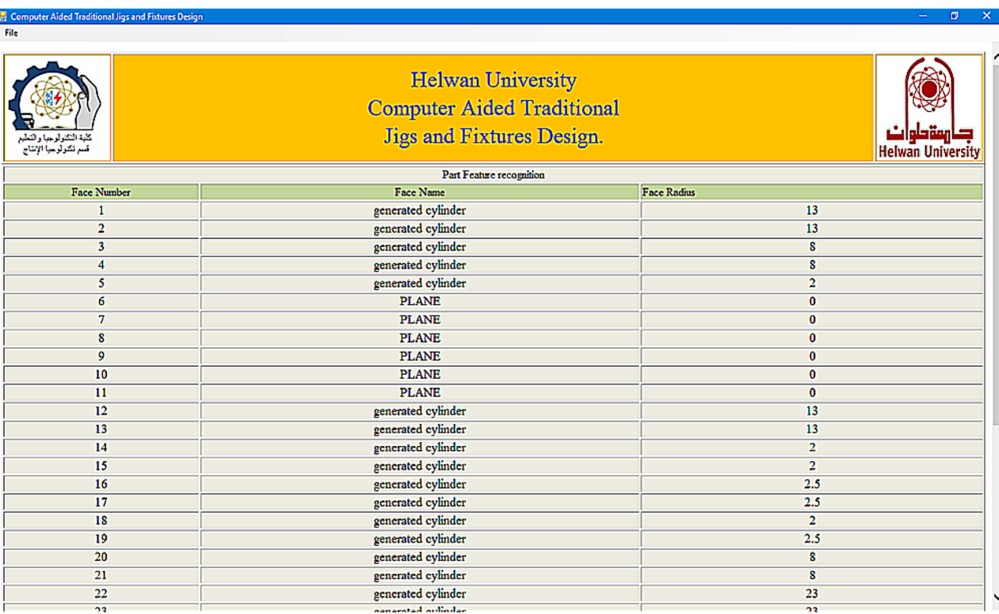

**Figure 12.** A report of the extracted data.

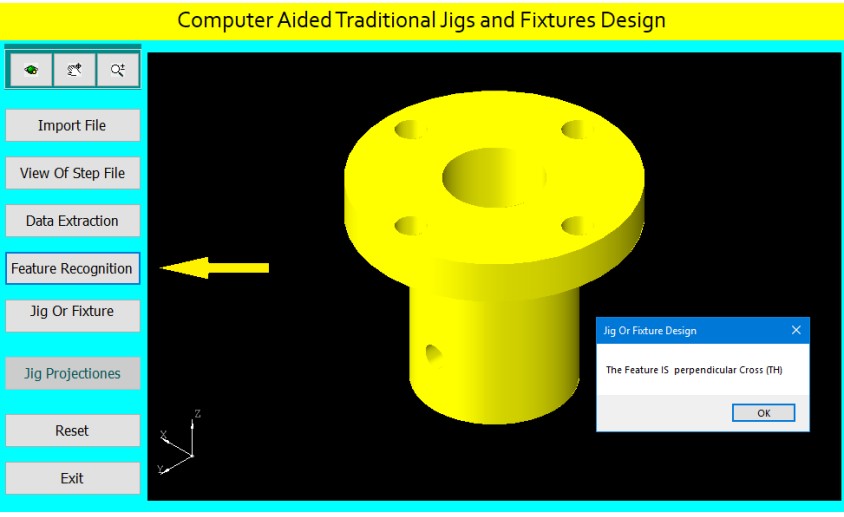

**Figure 13.** The fourth screen of the system interface.

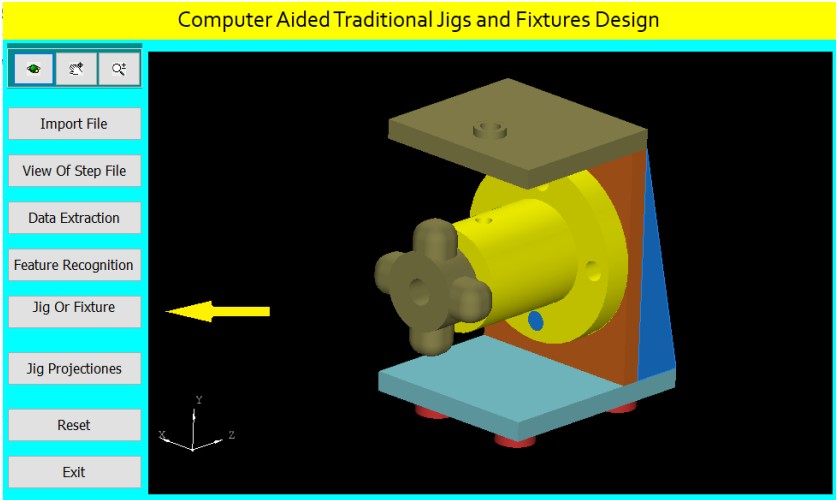

**Figure 14.** The fifth screen of the system interface.

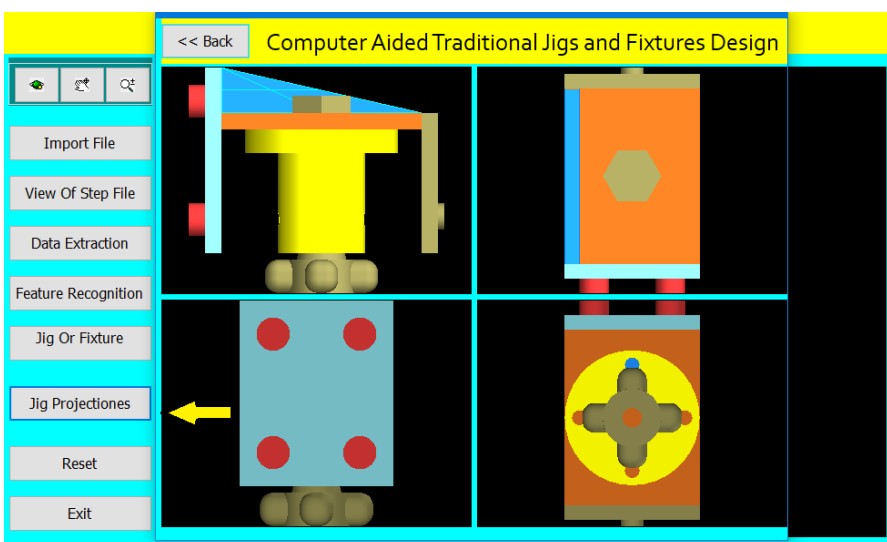

**Figure 15.** The sixth screen of the system interface.

*5.7. The Sixth Screen*

A Jig Design for a Cylindrical Part with a Perpendicular Cross Blind Hole

Figure 16 shows the drawing of the cylindrical part with a perpendicular cross blind hole.

Figure 17 shows the main components of the required jig design which will be done through the following steps:

The drawing and isometric of a cylindrical part were prepared using SolidWorks software as shown in Figure 16. The file is saved in STEP-AP203 file format.

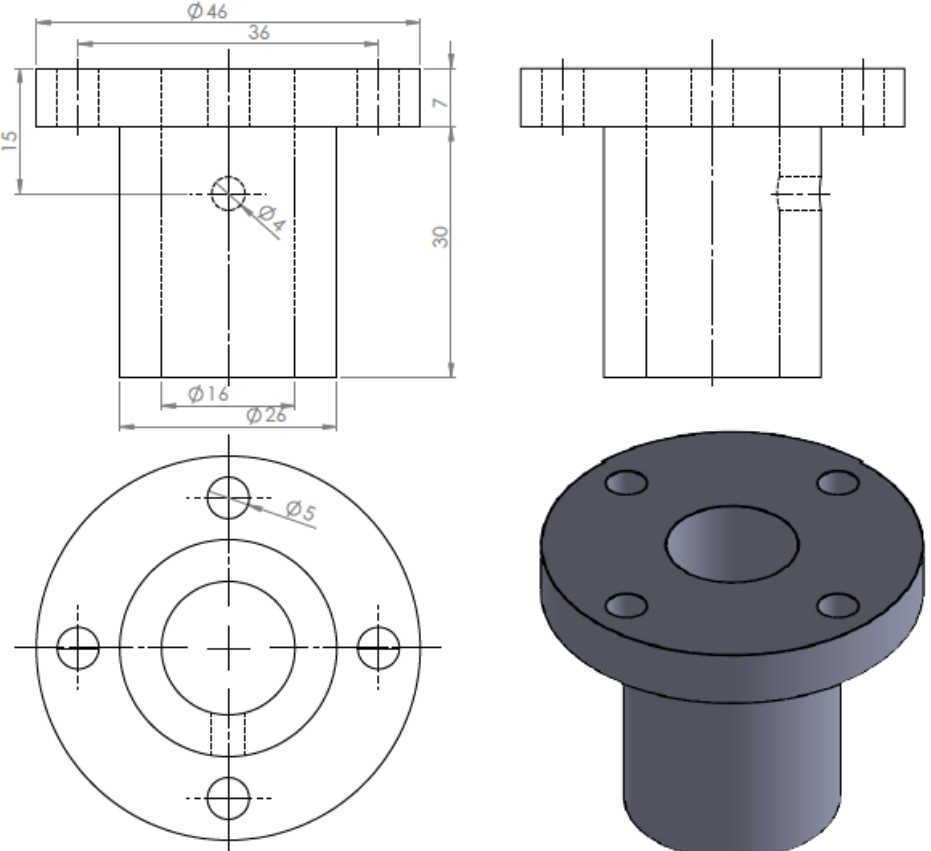

**Figure 16.** The drawing and isometric of the part.

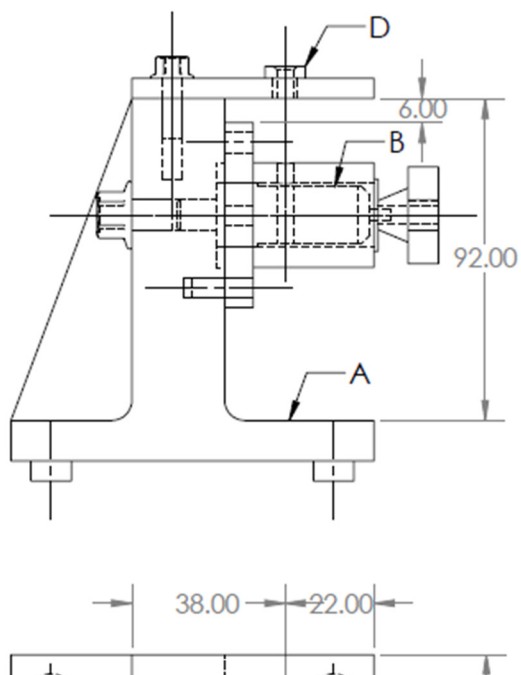
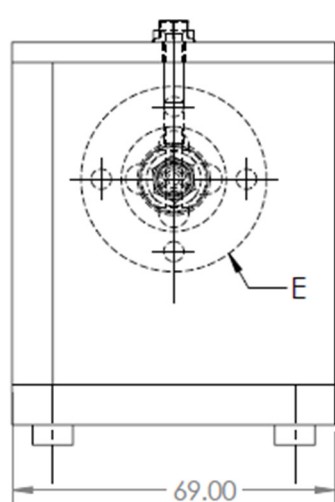
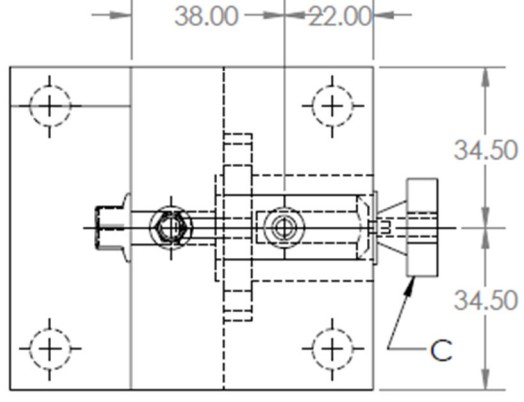

| A | Jig Body |
|---|---|
| B | Locator |
| C | Clamp |
| D | Bush Guide |
| E | Workpiece |

**Figure 17.** A drawing for the required jig design of the part.

The tailor-made software that is installed on the Visual Basic platform is used to extract the part data from the STEP AP-203 file and compare the data with the rules of feature recognition by searching for B-Spline-Curve-with Knots string; then, it counts the number of the BSp strings to identify the end condition of the cross-hole (eight strings), which indicates the cross-hole is blind. Then, the system reads the values of the vertex (1 and 2) of BSp which indicates that the cross-hole is not offset from the cylinder axis, then the system read the directions (Dx, Dy, and Dz) of BSp which identifies that the cross-hole is not inclined, and this means that the cross-hole is a perpendicular cross blind hole.

In the system interface, once the file is selected and imported, the system activates all the interface buttons and shows the part on its interface screen. Moreover, the system allows zooming in and out, dragging and dropping, and rotating the part.

In Figure 18, the system divides the interface screen into two parts to show the part with STEP AP-203 file, the highlighted string in the STEP AP-203 file is (B_Spline_Curve_With_Knots) which indicates is the existence of a cross-hole on the part body.

Figure 19 illustrates a report of the extracted data of the part. The system creates a geometrical data report of all the part features according to the data extraction rules, and this report helps in feature recognition and design of traditional jigs and fixtures for the cylindrical part.

The faces in the extracted report are shown on the 3D part model in Figure 20.

The software is also recognizing the perpendicular cross blind hole feature, as shown in Figure 21, by applying the rules to recognize the cross-hole feature and showing a message box of the feature recognitions process.

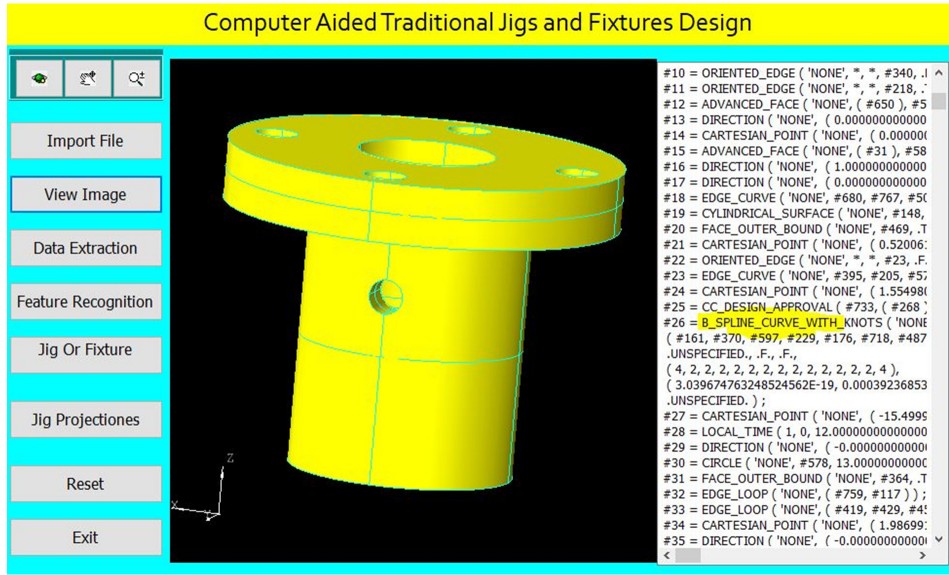

**Figure 18.** STEP-AP203 file of the part.

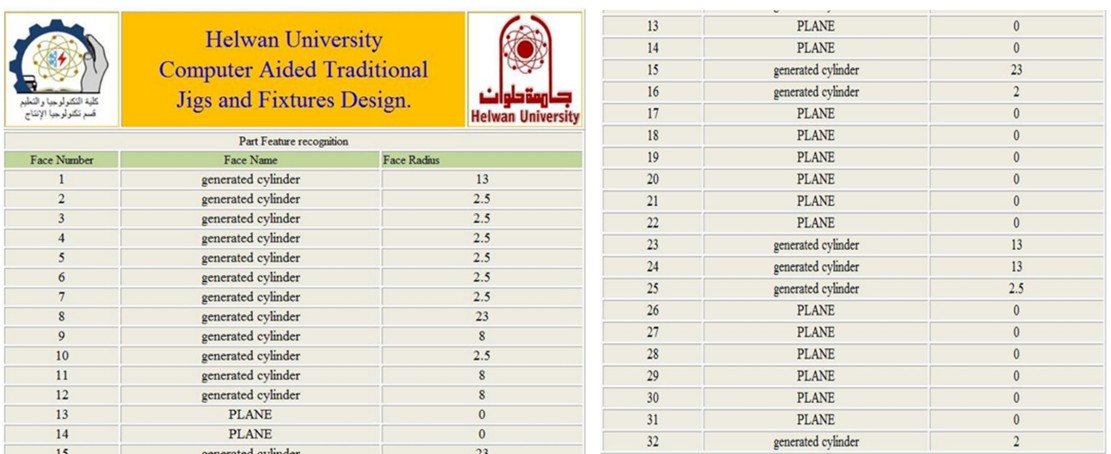

**Figure 19.** Report of the extracted data.

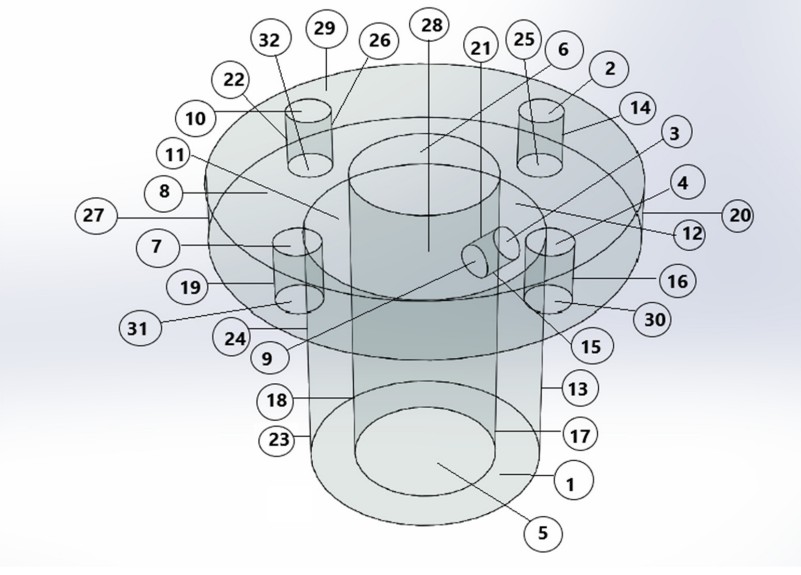

**Figure 20.** The faces of the part.

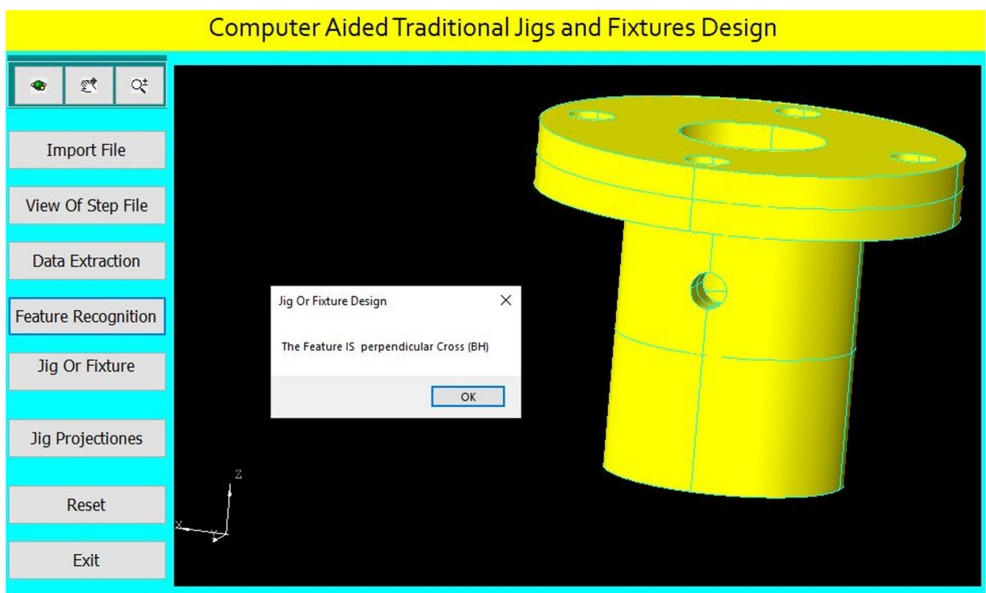

**Figure 21.** Automatic feature recognition of a perpendicular cross blind hole.

For drilling the perpendicular cross blind hole of (4 mm) diameter, the program checks the inner boring of (16 mm) diameter for identifying the appropriate locator diameter.

The program checks the perpendicular cross blind hole of (4 mm) diameter for identifying the suitable guide bush.

The program identifies the jig height according to the part maximum diameter as follows [31]:

$$H = 2Dmax$$

where (Dmax) is the maximum part diameter.

The program determines the center of the guide bush to be collinear with the cross-hole center as follows:

The cross-hole center = (x, y1, z)

The guide bush center = (x, y2, z)

The program identifies suitable clamp devices.

The jig design for this part is done parametrically on the software screen as shown in Figure 22.

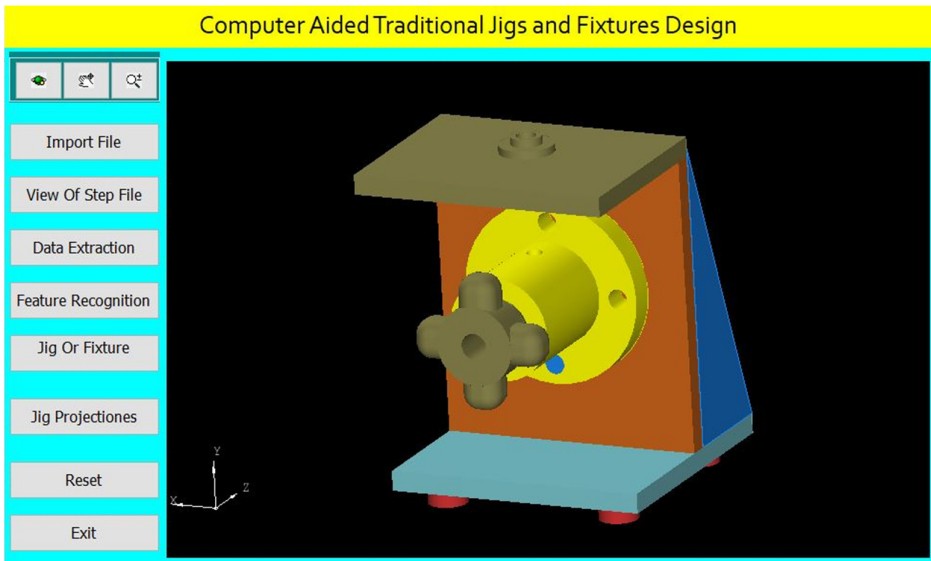

**Figure 22.** Computer-aided design of a jig for the cylindrical part.

The software shows the jig views in Figure 23. The button (Jig Projections) is used to draw the jigs views on the interface screen to show the details of the design of traditional jigs and fixtures.

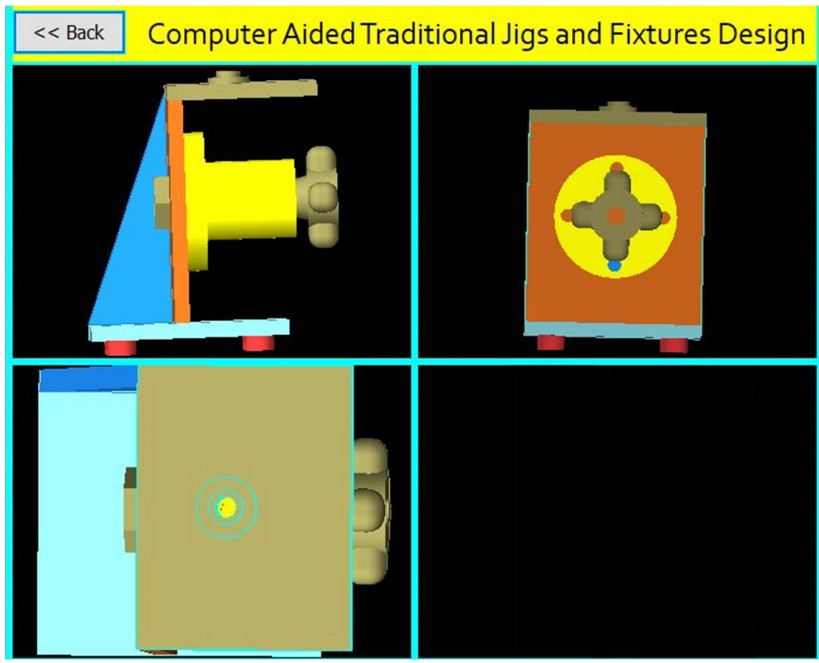

**Figure 23.** The jig views.

The first case was explained in detail, but the following seven cases will be explained briefly:

A Jig Design for a Cylindrical Part with an Offset Cross Blind Hole

Figure 24 shows the drawing of the cylindrical part with an offset cross blind hole, and Figure 25 shows the main components of the required jig design which will be done through the following steps:

The drawing and isometric of a cylindrical part prepared using SolidWorks software as shown in Figure 24. The file is saved in STEP-AP203 file format.

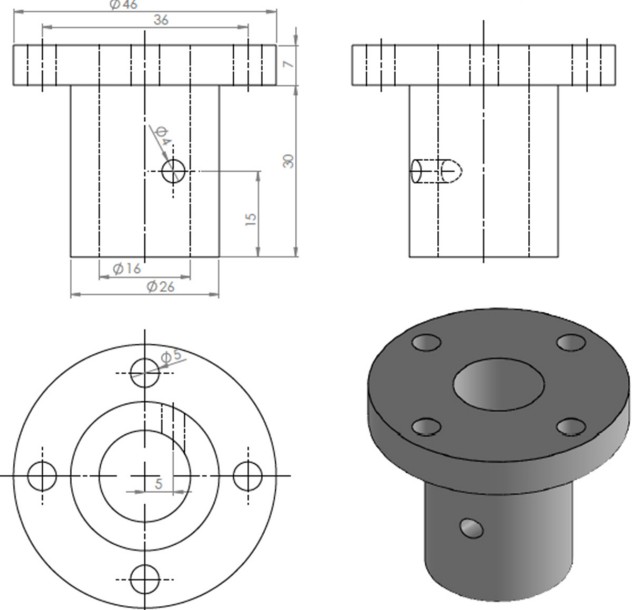

**Figure 24.** The drawing and isometric of the part.

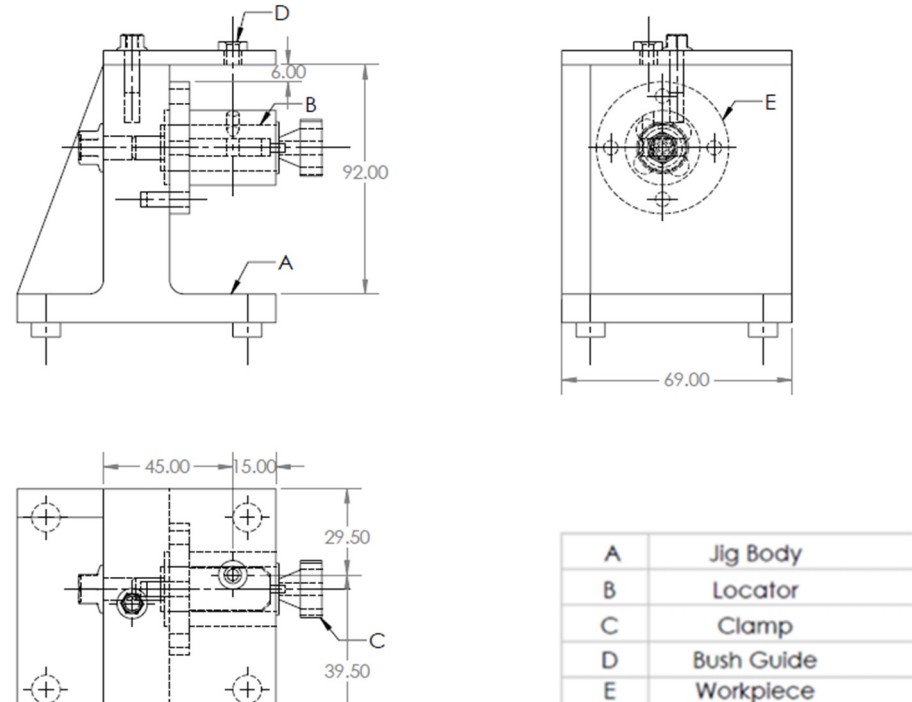

| A | Jig Body |
|---|----------|
| B | Locator |
| C | Clamp |
| D | Bush Guide |
| E | Workpiece |

**Figure 25.** A drawing for the required jig design of the part.

The tailor-made software which is installed on the Visual Basic platform is used to extract the part data from the STEP AP-203 file and compare these data with the rules of feature recognition by searching for B-Spline-Curve-with Knots string; then, it counts the number of the BSp strings to identify the end condition of the cross-hole (eight strings), which identifies that the cross-hole is blind. Then, the system reads the values of the vertex (1 and 2) of BSp and both values are ($\neq 0$ mm) which indicates that the cross-hole is offset from the axis of the cylinder, and then the system reads the directions (Dx, Dy, and Dz) of BSp which identifies that the cross-hole is not inclined, which means that the cross-hole is an offset cross blind hole.

The software is also recognizing the offset cross blind hole feature as shown in Figure 26 by applying the rules to recognize the cross-hole feature and showing a message box of the feature recognitions process.

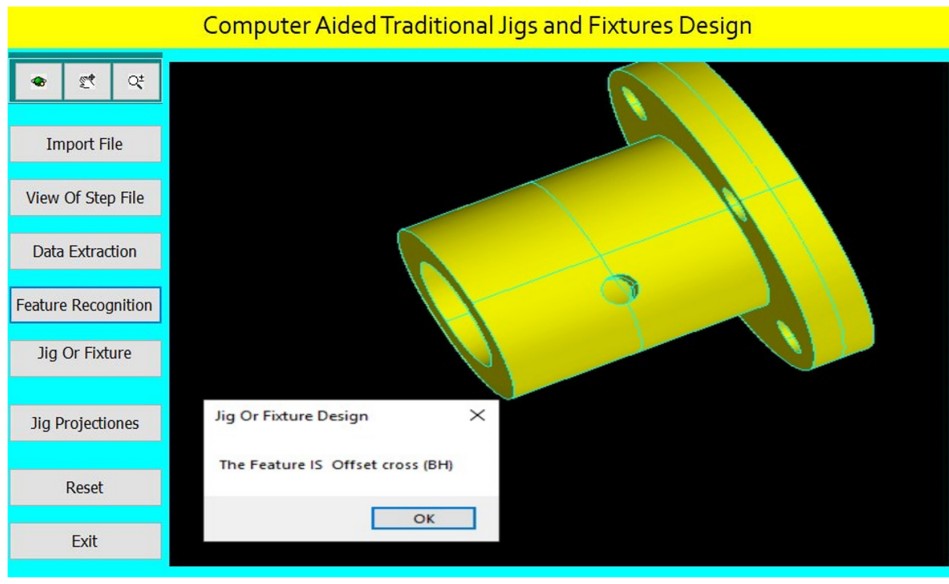

**Figure 26.** Automatic feature recognition of an offset cross blind hole.

The program checks the offset cross blind hole of (4 mm) diameter for identifying the suitable guide bush.

The design of the jig for this part is done parametrically on the software screen as shown in Figure 27.

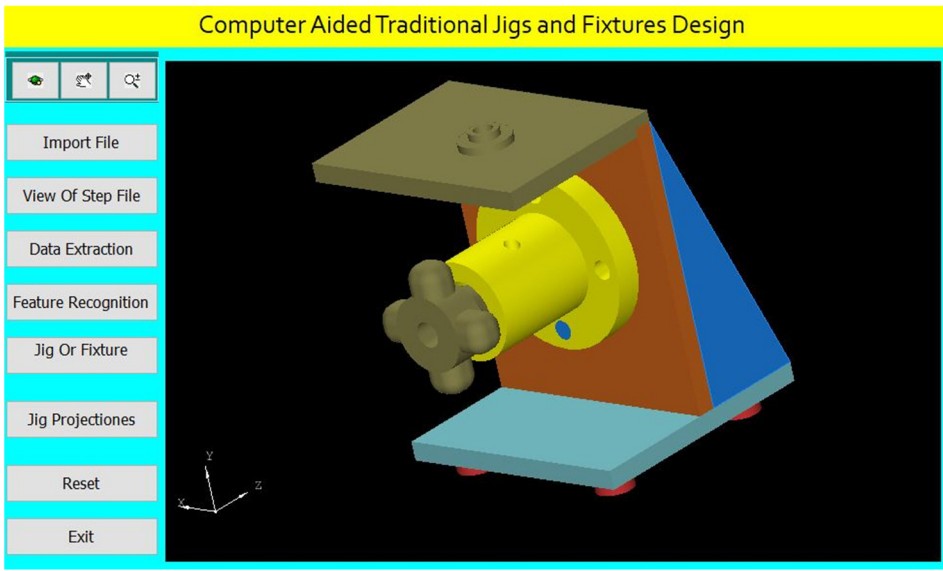

**Figure 27.** Computer-aided design of a jig for the cylindrical part.

The software shows the jig views in Figure 28. The button (Jig Projections) is used to draw the jigs views on the interface screen to show the details of the design of traditional jigs and fixtures.

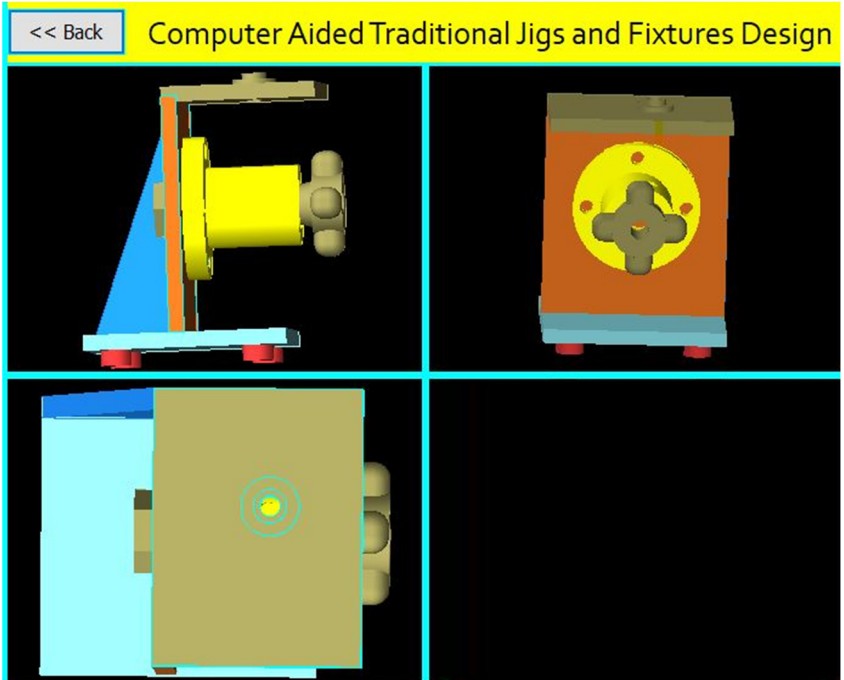

**Figure 28.** The jig views.

A Jig Design for a Cylindrical Part with an Inclined Cross Blind Hole.

Figure 29 shows the drawing of the cylindrical part with an inclined cross blind hole. Figure 30 shows the main components of the required jig design which will be done through the following steps:

The drawing and isometric of a cylindrical part were prepared using SolidWorks software as shown in Figure 29. The file is saved in STEP-AP203 file format.

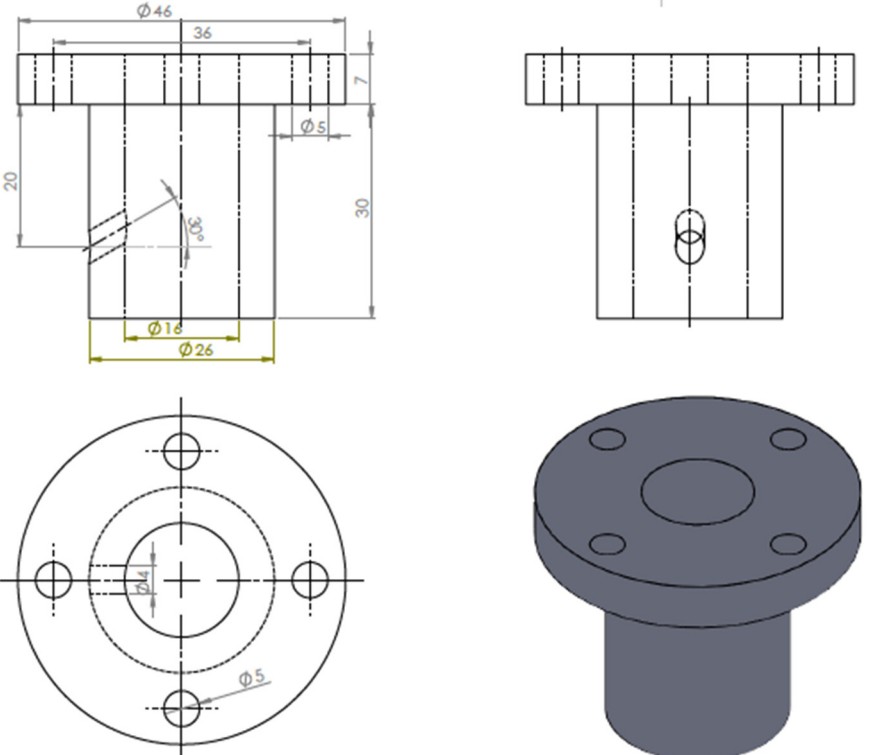

**Figure 29.** The drawing and isometric of the part.

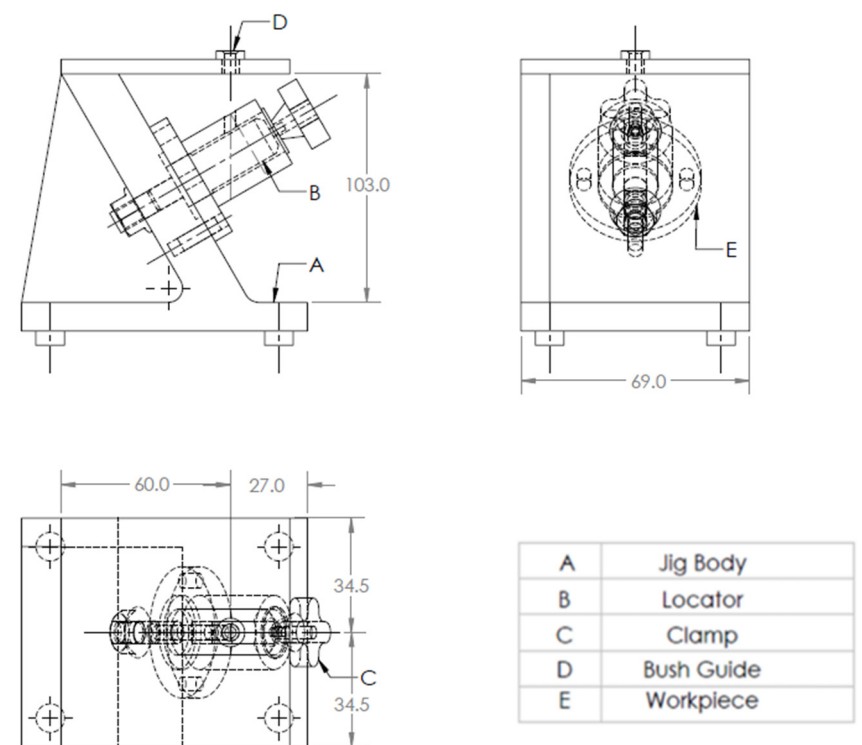

| | |
|---|---|
| A | Jig Body |
| B | Locator |
| C | Clamp |
| D | Bush Guide |
| E | Workpiece |

**Figure 30.** A drawing for the required jig design of the part.

The tailor-made software which is installed on the Visual Basic platform is used to extract the part data from the STEP AP-203 file and compare these data with the rules of feature recognition by searching for B-Spline-Curve-with Knots string; then, it counts the

number of the BSp strings to identify the end condition of the cross-hole (eight strings), which illustrates the cross-hole is blind. Then, the system reads the values of the vertex (1 and 2) of BSp which indicates that the cross-hole is not offset from the axis of the cylinder. Then, the system reads the directions (Dx, Dy, and Dz) of BSp and the values of two directions of these three are (>0 mm), which identifies that the cross-hole is inclined, and this means that the cross-hole is an inclined cross blind hole.

The software is also recognizing the inclined cross blind hole feature as shown in Figure 31 by applying the rules to recognize the cross-hole feature and showing a message box of the feature recognitions process.

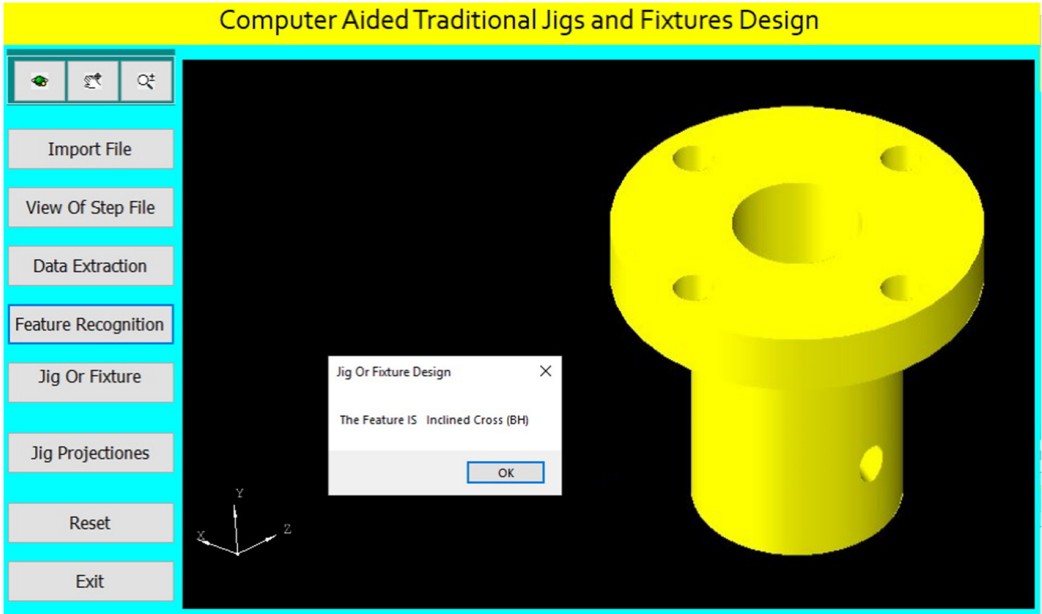

**Figure 31.** Automatic feature recognition of an inclined cross blind hole.

The program checks the inclined cross blind hole of (4 mm) diameter for identifying the suitable guide bush.

The program identifies the jig height according to the part maximum diameter of the part and the inclination angle of the cross-hole as follows:

$$H = 2Dmax/\cos (\theta \text{ rad})$$

$$\theta \text{ rad} = \theta° \pi/180$$

where (Dmax) is the maximum diameter of the part and ($\theta°$) is the inclination angle of the cross-hole.

The design of the jig for this part is done parametrically on the software screen as shown in Figure 32.

The software shows the jig views in Figure 33. The button (Jig Projections) is used to draw the jigs views on the interface screen to show the details of the design of traditional jigs and fixtures.

A Jig Design for a Cylindrical Part with an Inclined Offset Cross Blind Hole

Figure 34 shows the drawing of the cylindrical part with an inclined offset blind hole, and Figure 35 shows the main components of the required jig design which will be done through the following steps:

The drawing and isometric of a cylindrical part were prepared using SolidWorks software as shown in Figure 34. The file is saved in STEP-AP203 file format.

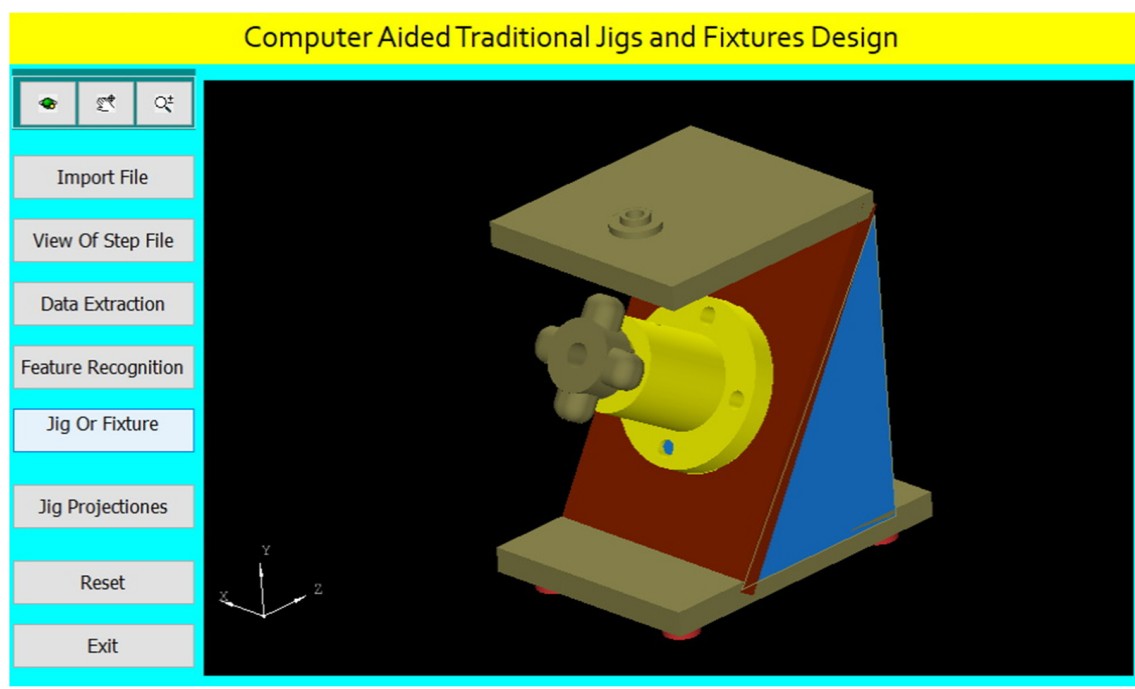

**Figure 32.** Computer-aided design of a jig for the cylindrical part.

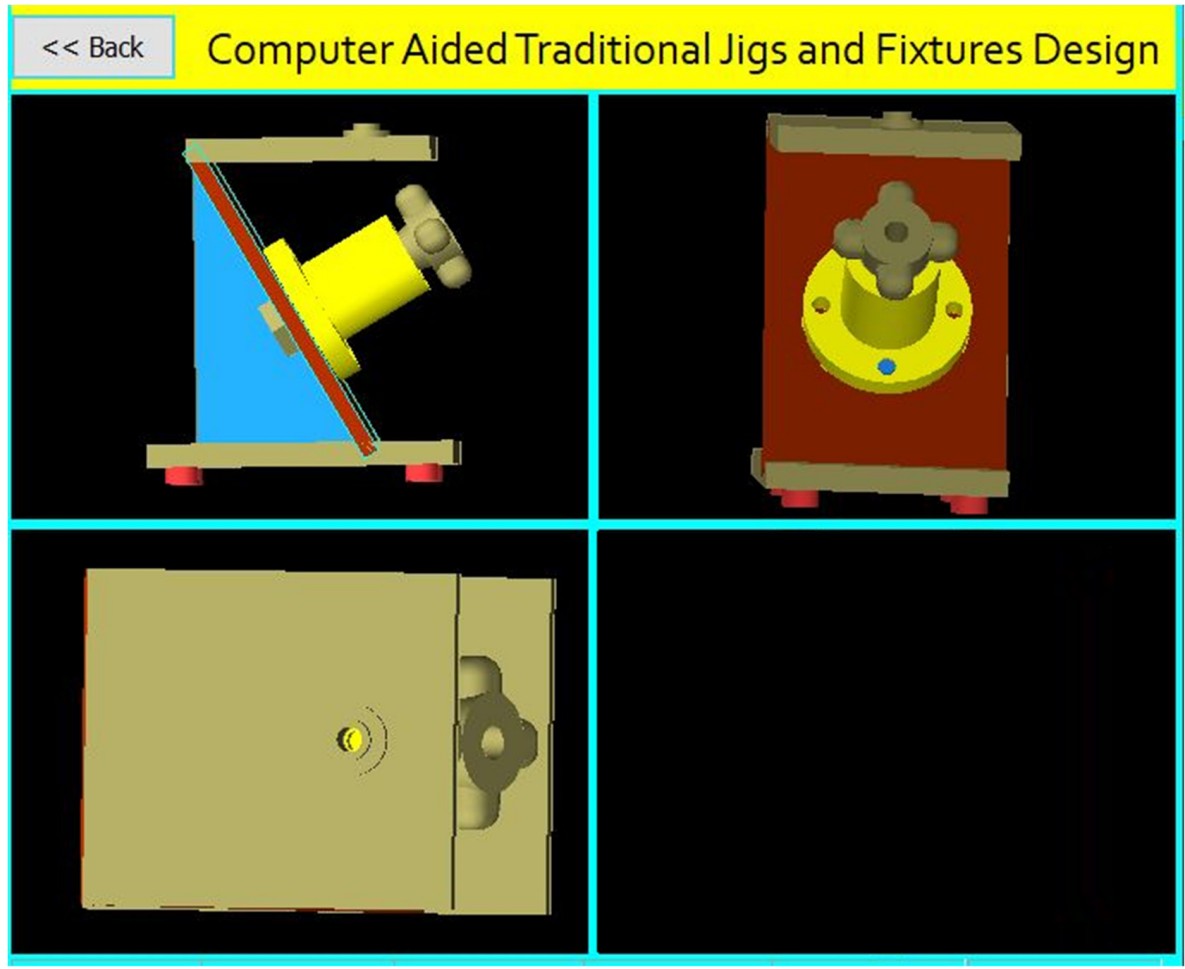

**Figure 33.** The jig views.

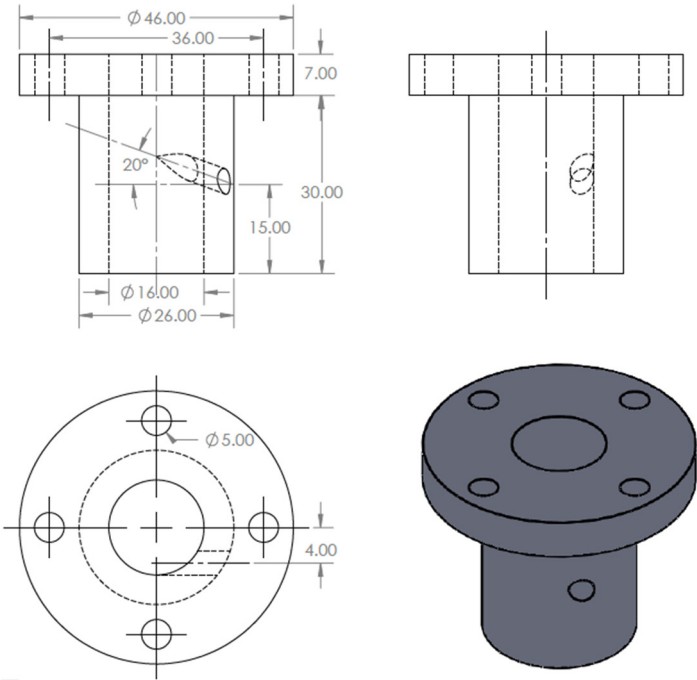

**Figure 34.** The drawing and isometric of the part.

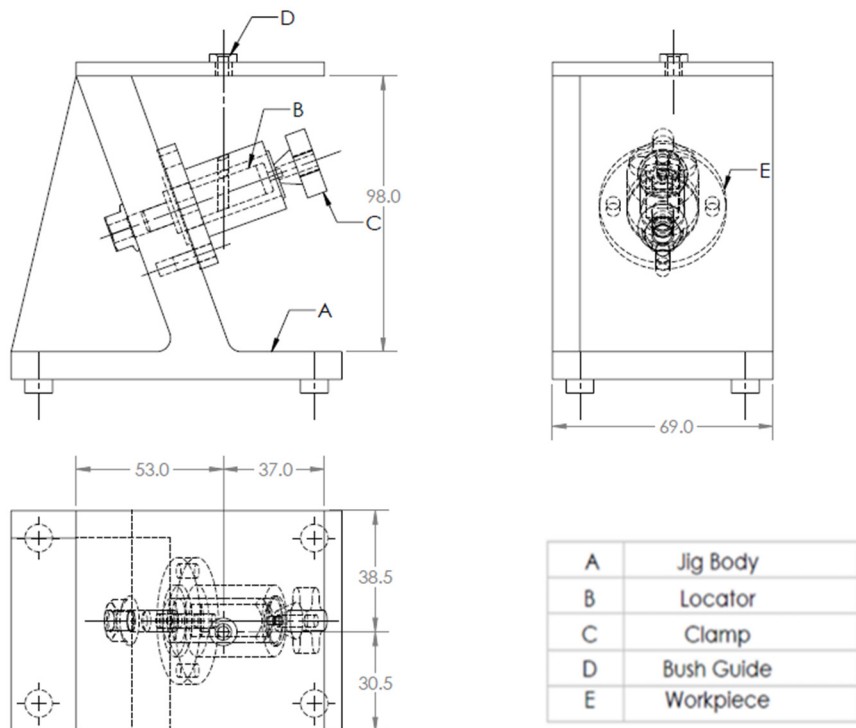

| A | Jig Body |
|---|----------|
| B | Locator |
| C | Clamp |
| D | Bush Guide |
| E | Workpiece |

**Figure 35.** A drawing for the required jig design of the part.

The tailor-made software which is installed on the Visual Basic platform is used to extract the part data from the STEP AP-203 file and compare these data with the feature recognition rules by searching for B-Spline-Curve-with Knots string; then, it counts the number of the BSp strings to identify the end condition of the cross-hole (eight strings), which indicates the cross-hole is blind. Then, the system reads the values of the vertex (1 and 2) of BSp and both values are ($\neq 0$ mm) which indicates that the cross-hole is offset from the axis of the cylinder. The system reads the directions (Dx, Dy, and Dz) of BSp, and

the values of two directions of these three are (>0 mm), which identifies that the cross-hole is inclined, and this means that the cross-hole is an inclined offset cross blind hole.

The software is also recognizing the inclined offset cross blind hole feature as shown in Figure 36 by applying the rules to recognize the cross-hole feature and showing a message box of the feature recognitions process.

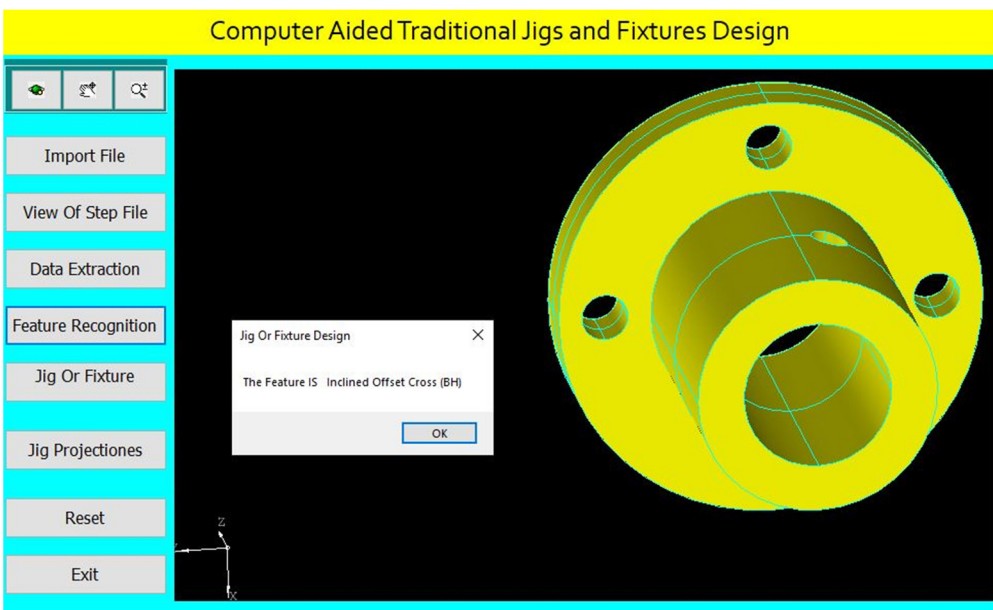

**Figure 36.** Automatic feature recognition of an inclined offset cross blind hole.

The program checks the inclined offset cross blind hole of (4 mm) diameter for identifying the suitable guide bush.

The program identifies suitable clamp devices.

The design of the jig for this part is done parametrically on the software screen as shown in Figure 37.

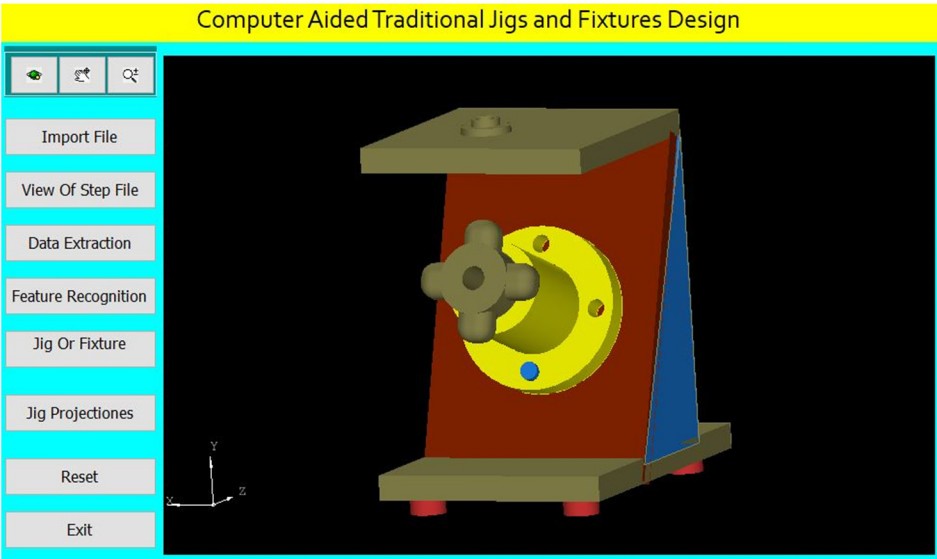

**Figure 37.** Computer-aided design of a jig for the cylindrical part.

The software shows the jig views in Figure 38. The button (Jig Projections) is used to draw the jigs views on the interface screen to show the details of the design of traditional jigs and fixtures.

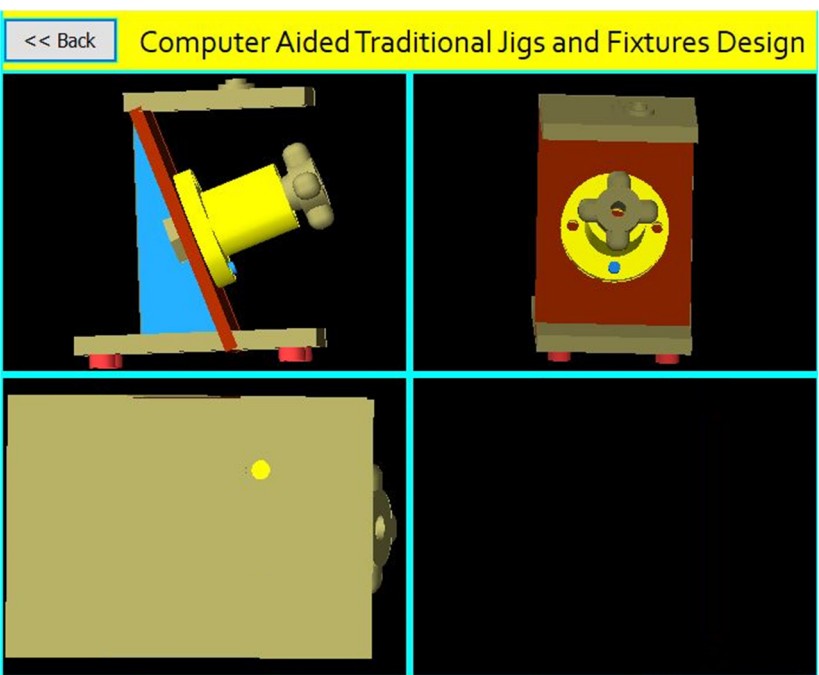

**Figure 38.** The jig views.

A Jig Design for a Cylindrical Part with a Perpendicular Cross Through-Hole

Figure 39 shows the drawing of the cylindrical part with a perpendicular cross-through-hole, and Figure 40 shows the main components of the required jig design which will be done through the following steps:

The drawing and isometric of a cylindrical part were prepared using SolidWorks software as shown in Figure 39. The file is saved in STEP-AP203 file format.

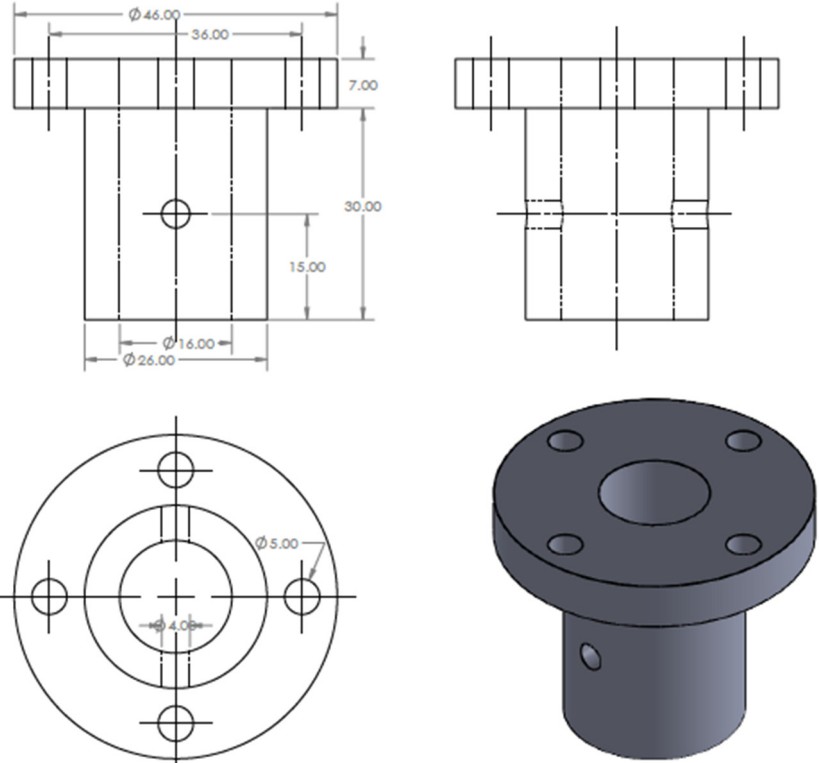

**Figure 39.** The drawing and isometric of the part.

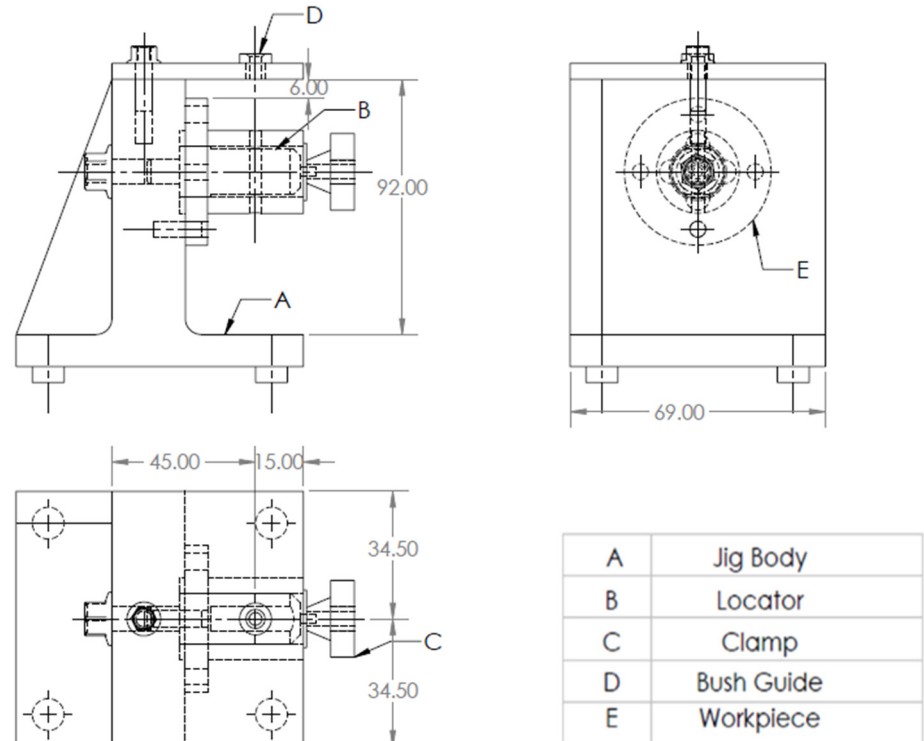

**Figure 40.** A drawing for the required jig design of the part.

The tailor-made software which is installed on the Visual Basic platform is used to extract the part data from the STEP AP-203 file and compare these data with the rules of feature recognition by searching for B-Spline-Curve-with Knots string; then, it counts the number of the BSp strings to identify the end condition of the cross-hole (16 strings) which identifies the cross-hole is through. Then, the system reads the values of the vertex (1 and 2) of BSp which indicates that the cross-hole is not offset from the axis of the cylinder; the system also reads the directions (Dx, Dy, and Dz) of BSp which identifies that the cross-hole is not inclined, and this means that the cross-hole is a perpendicular cross through-hole.

The software is also recognizing the perpendicular cross through-hole feature as shown in Figure 41 by applying the rules to recognize the cross-hole feature and showing a message box of the feature recognitions process.

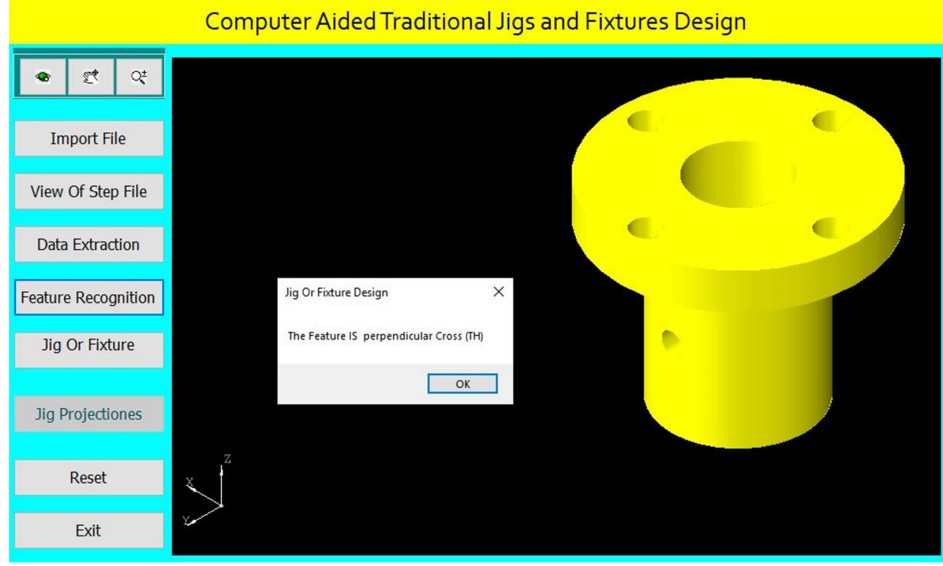

**Figure 41.** Automatic feature recognition of a perpendicular cross through-hole.

The program checks the perpendicular cross through-hole of (4 mm) diameter for identifying the suitable guide bush.

The design of the jig for this part is done parametrically on the software screen as shown in Figure 42.

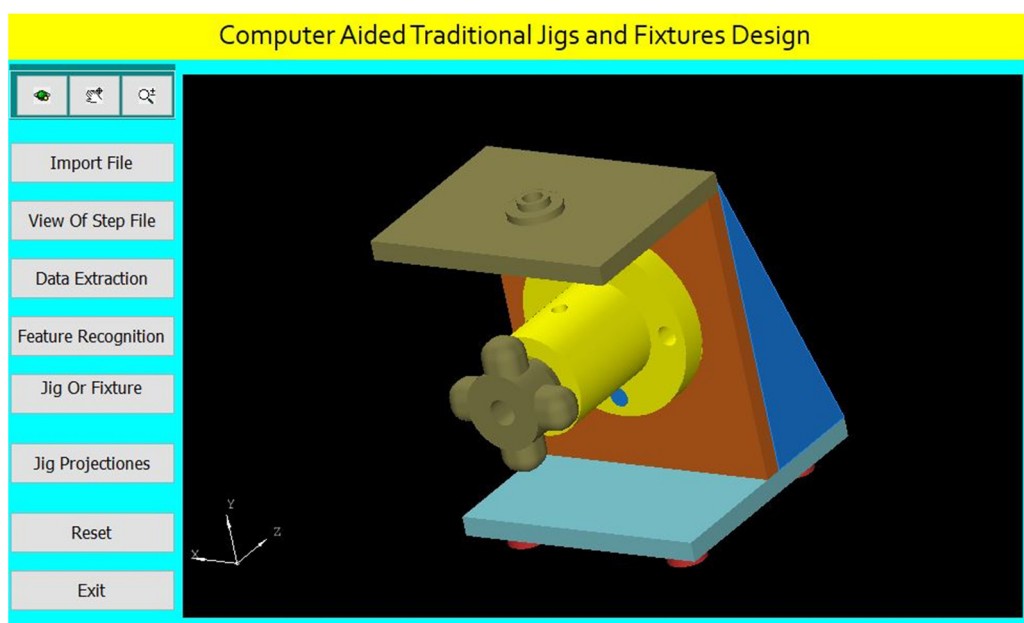

**Figure 42.** Computer-aided design of a jig for the cylindrical part.

The software shows the jig views in Figure 43. The button (Jig Projections) is used to draw the jigs views on the interface screen to show the details of the design of traditional jigs and fixtures.

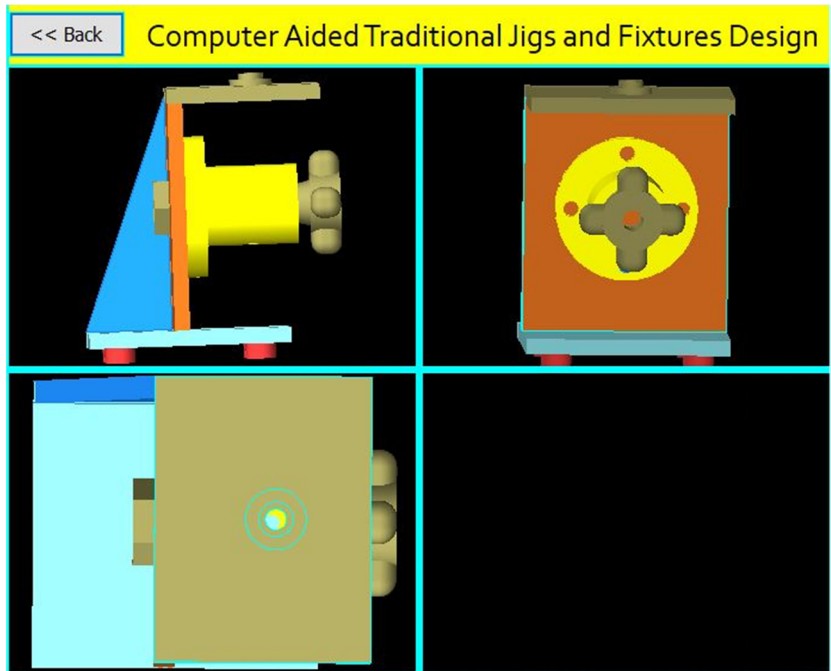

**Figure 43.** The jig views.

A Jig Design for a Cylindrical Part with an Offset Cross Through-Hole.

Figure 44 Shows the drawing of the cylindrical part with an offset cross through-hole, while Figure 45 shows the main components of the required jig design, which will be done through the following steps:

The drawing and isometric of a cylindrical part were prepared using SolidWorks software as shown in Figure 44. The file is saved in STEP-AP203 file format.

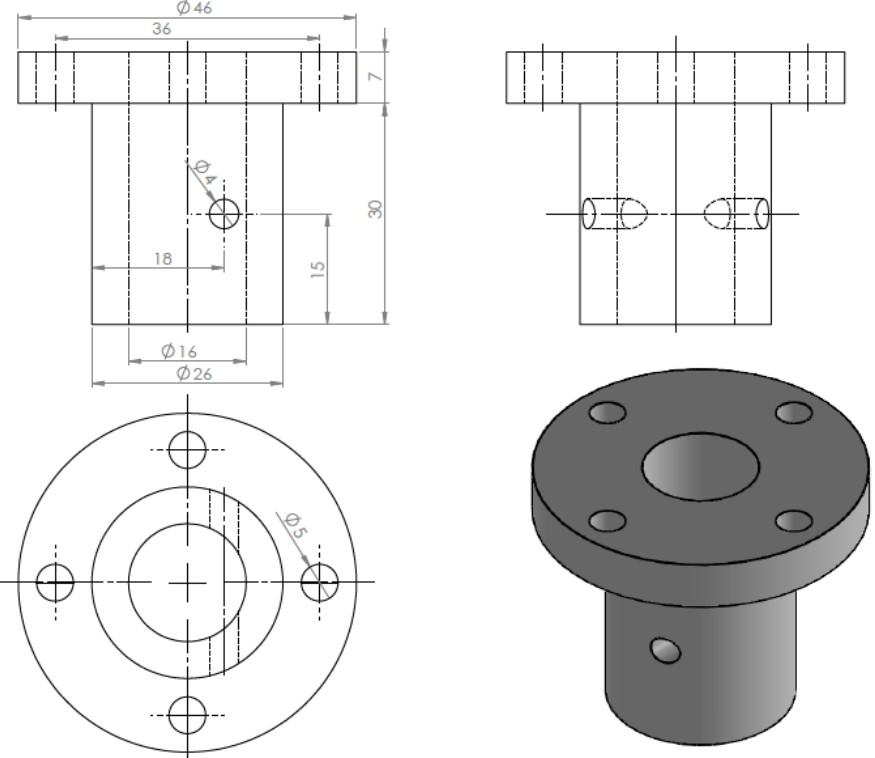

**Figure 44.** The drawing and isometric of the part.

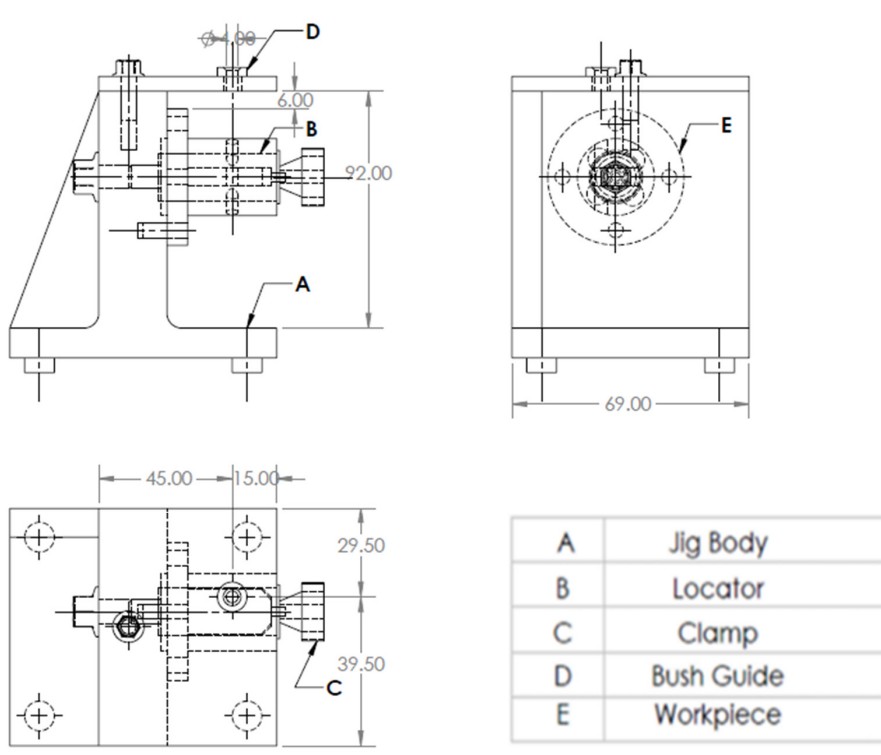

| A | Jig Body |
|---|---|
| B | Locator |
| C | Clamp |
| D | Bush Guide |
| E | Workpiece |

**Figure 45.** A drawing for the required jig design of the part.

The tailor-made software which is installed on the Visual Basic platform is used to extract the part data from the STEP AP-203 file and compare these data with the rules of feature recognition by searching for B-Spline-Curve-with Knots string; then, it counts the number of the BSp strings to identify the end condition of the cross-hole (16 strings) which illustrates the cross-hole is through. Then, the system reads the values of the vertex (1 and 2) of BSp, and both values are ($\neq$0 mm), which indicates that the cross-hole is offset from the axis of the cylinder. Then, the system reads the directions (Dx, Dy, and Dz) of BSp which identifies that the cross-hole is not inclined, and this means that the cross-hole is an offset cross through-hole.

The software is also recognizing the offset cross through-hole feature as shown in Figure 46 by applying the rules to recognize the cross-hole feature and showing a message box of the feature recognitions process.

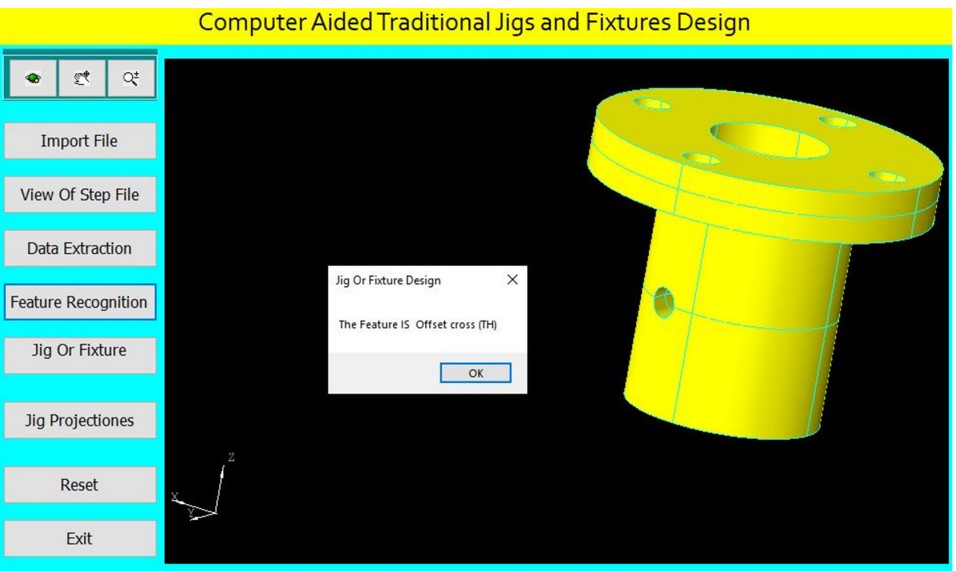

**Figure 46.** Automatic feature recognition of an offset cross through-hole.

The program checks the offset cross through-hole of (4 mm) diameter for identifying the suitable guide bush.

Design of the jig for this part is done parametrically on the software screen as shown in Figure 47.

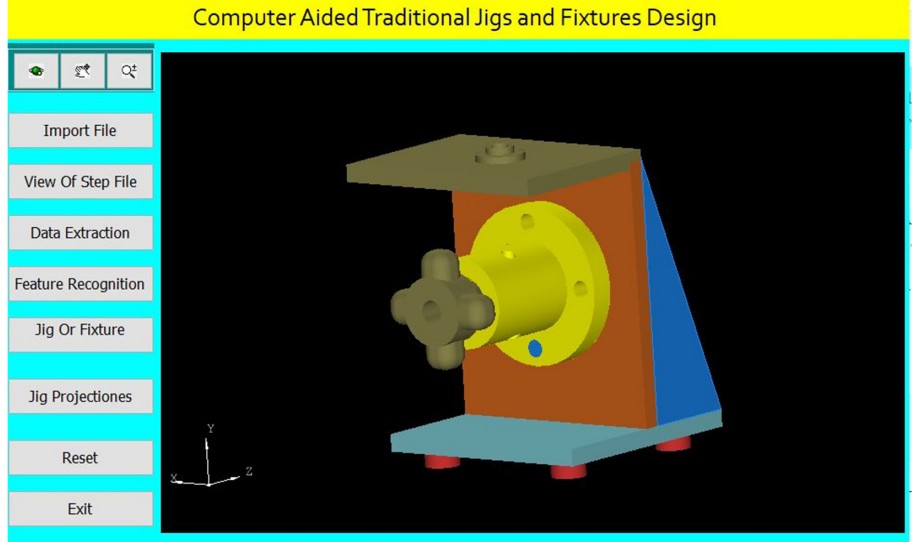

**Figure 47.** Computer-aided design of a jig for the cylindrical part.

The software shows the jig views in Figure 48. The button (Jig Projections) is used to draw the jigs views on the interface screen to show the details of the design of traditional jigs and fixtures.

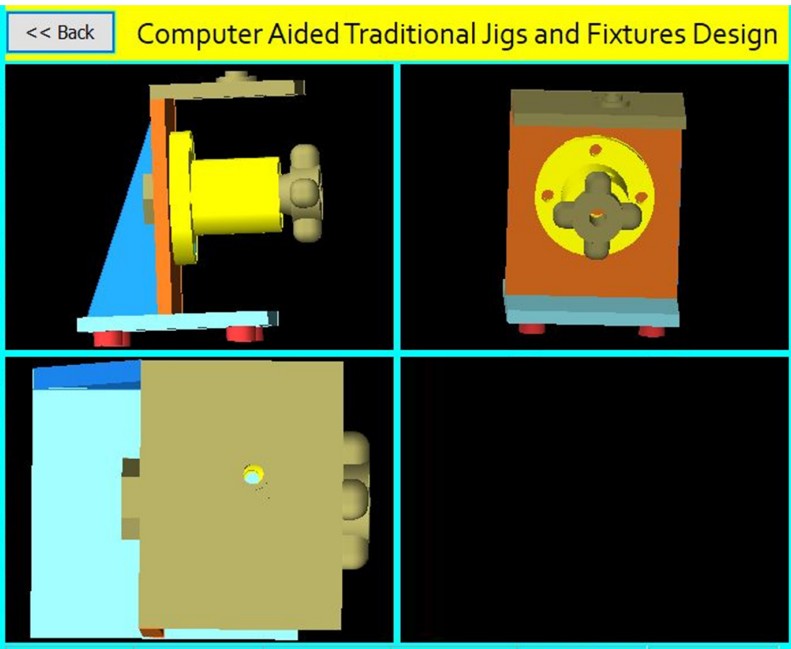

**Figure 48.** The jig views.

A Jig Design for a Cylindrical Part with an Inclined Cross Through-Hole.

Figure 49 shows the drawing of the cylindrical part with an inclined cross through-hole, and Figure 50 shows the main components of the required jig design which will be done through the following steps:

The drawing and isometric of a cylindrical part prepared using SolidWorks software as shown in Figure 49. The file is saved in STEP-AP203 file format.

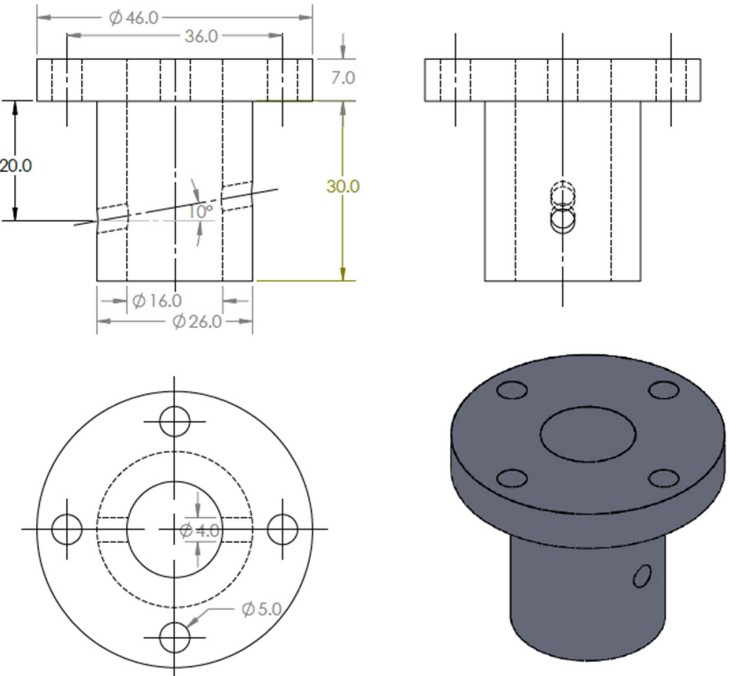

**Figure 49.** The drawing and isometric of the part.

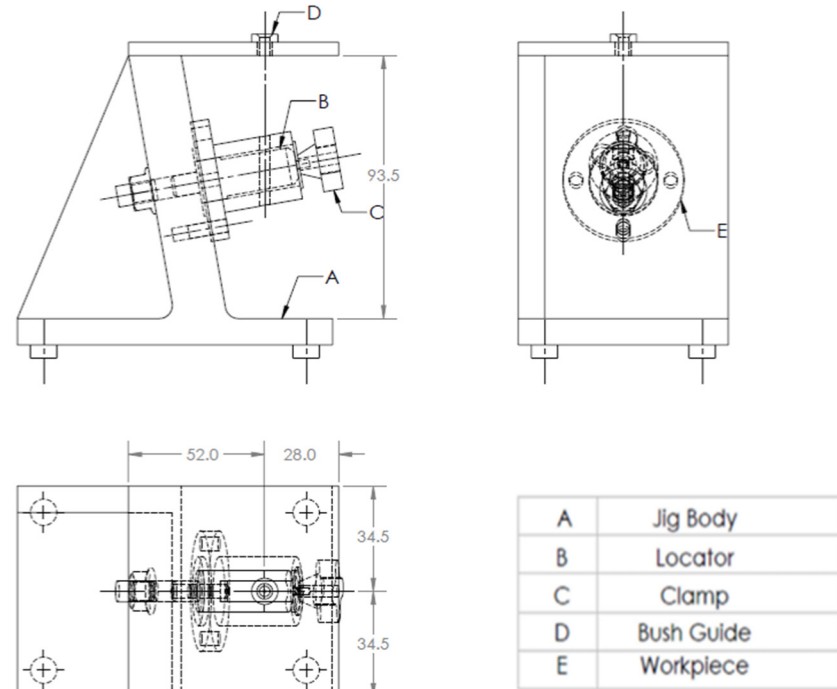

**Figure 50.** A drawing for the required jig design of the part.

The tailor-made software which is installed on the Visual Basic platform is used to extract the part data from the STEP AP-203 file and compare these data with the rules of feature recognition by searching for B-Spline-Curve-with Knots string; then, it counts the number of the BSp strings to identify the end condition of the cross-hole (16 strings) which indicates the cross-hole is through. Then, the system reads the values of the vertex (1 and 2) of BSp which indicates that the cross-hole is not offset from the axis of the cylinder, the system read the directions (Dx, Dy, and Dz) of BSp, and the values of two directions of these three are (>0 mm), which identifies that the cross-hole is inclined; this means that the cross-hole is an inclined cross through-hole.

The software is also recognizing the inclined cross through-hole feature as shown in Figure 51 by applying the rules to recognize the cross-hole feature and showing a message box of the feature recognitions process.

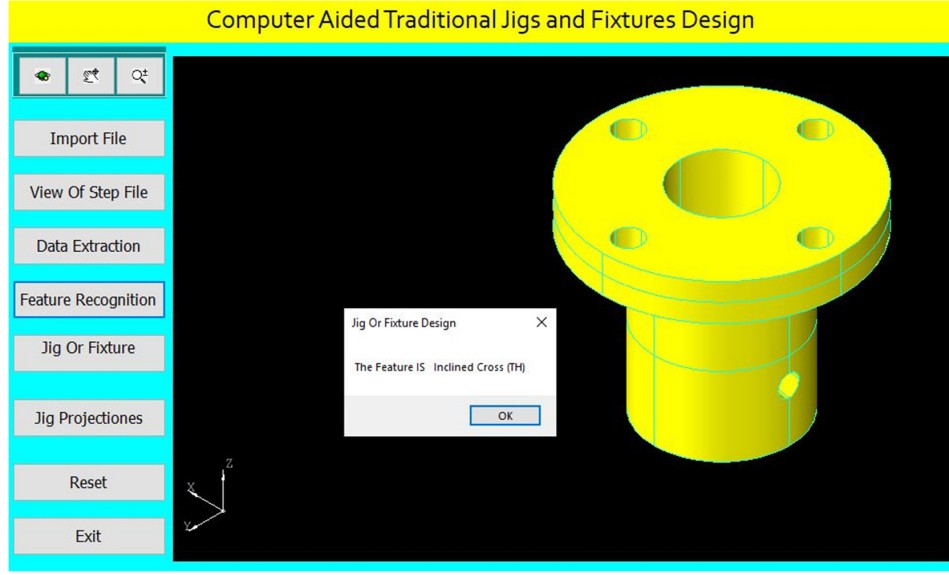

**Figure 51.** Automatic feature recognition of an inclined cross through-hole.

The program checks the inclined cross through-hole of (4 mm) diameter for identifying the suitable guide bush.

Design of the jig for this part is done parametrically on the software screen as shown in Figure 52.

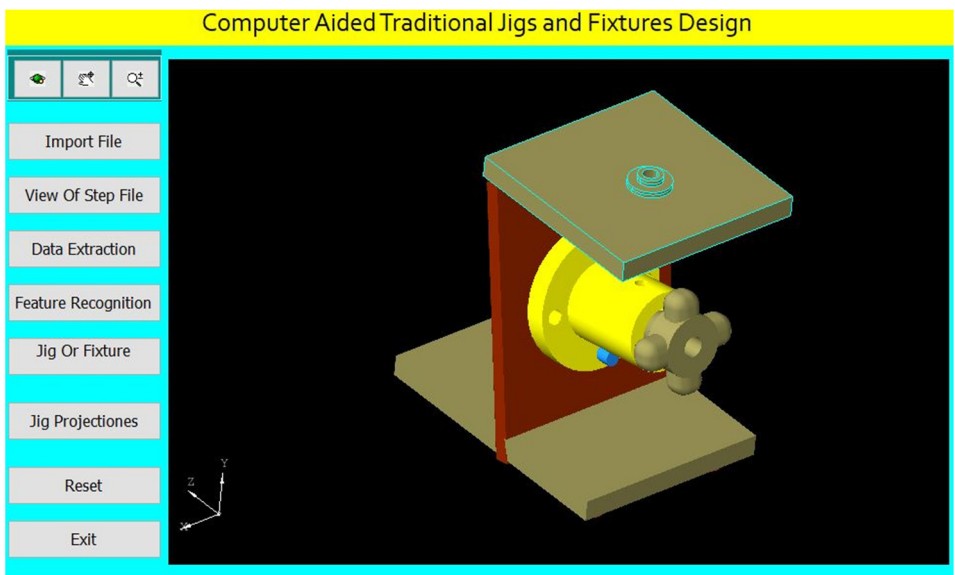

**Figure 52.** Computer-aided design of a jig for the cylindrical part.

The software shows the jig views in Figure 53. The button (Jig Projections) is used to draw the jigs views on the interface screen to show the details of the design of traditional jigs and fixtures.

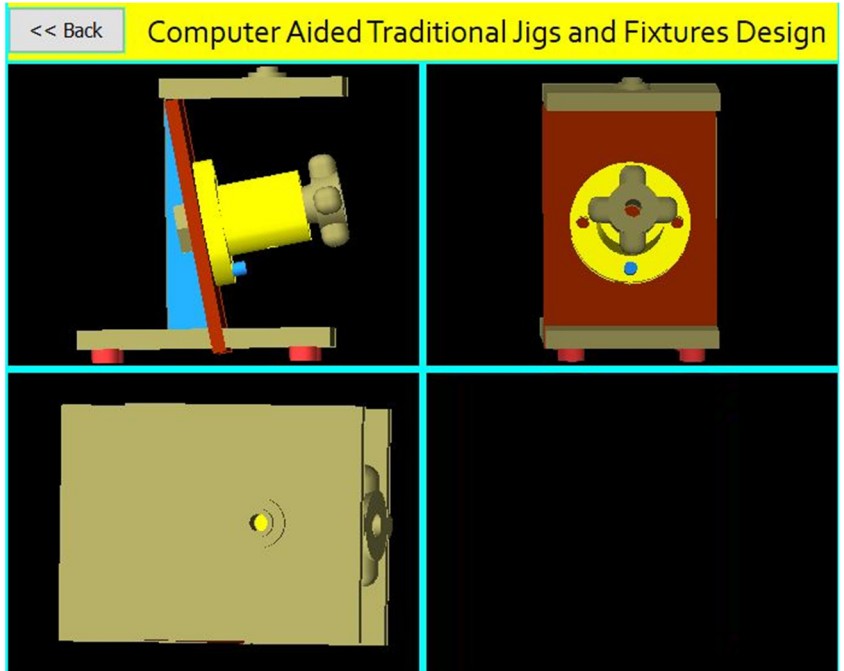

**Figure 53.** The jig views.

A Jig Design for a Cylindrical Part with an Inclined Offset Cross Through-Hole.

Figure 54 shows the drawing of the cylindrical part with an inclined offset through-hole. Figure 55 shows the main components of the required jig design which will be done through the following steps:

The drawing and isometric of a cylindrical part were prepared using SolidWorks software as shown in Figure 54. The file is saved in STEP-AP203 file format.

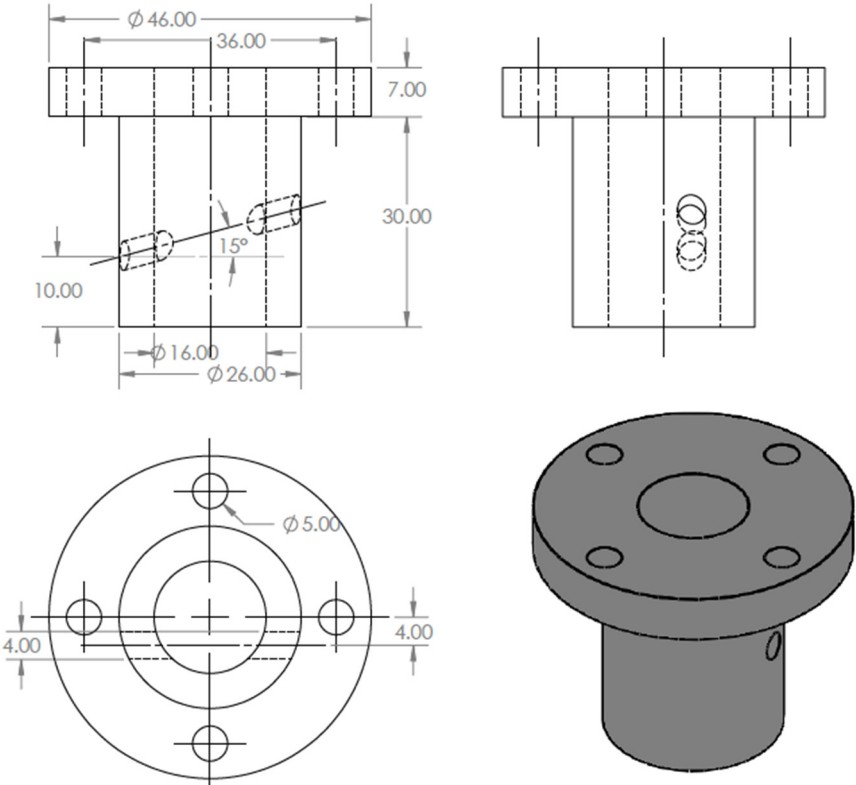

**Figure 54.** The drawing and isometric of the part.

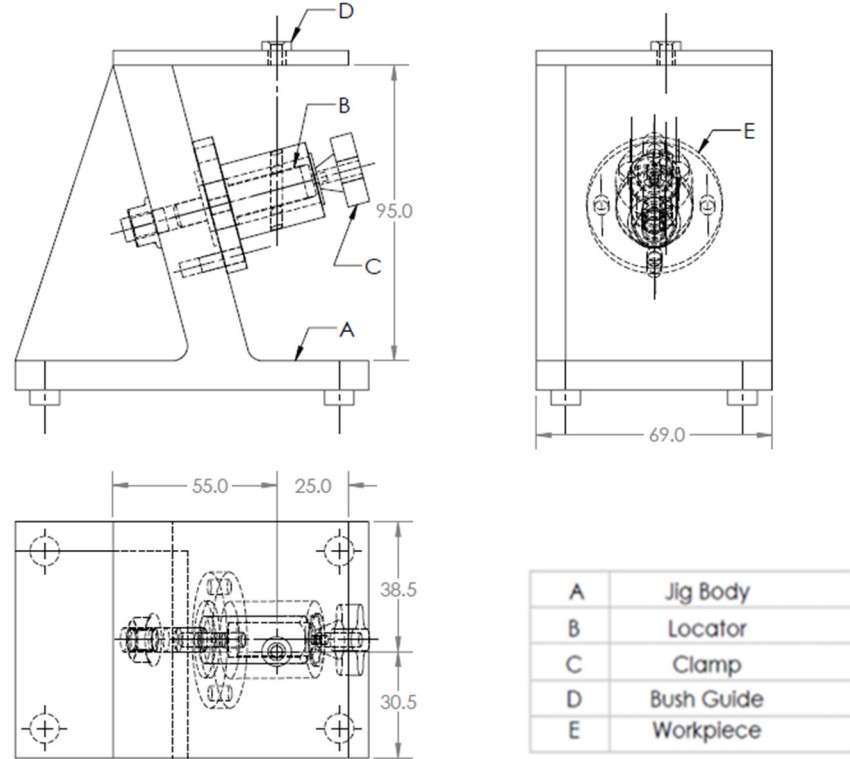

| | |
|---|---|
| A | Jig Body |
| B | Locator |
| C | Clamp |
| D | Bush Guide |
| E | Workpiece |

**Figure 55.** A drawing for the required jig design of the part.

The tailor-made software which is installed on the Visual Basic platform is used to extract the part data from the STEP AP-203 file and compare these data with the rules of feature recognition by searching for B-Spline-Curve-with Knots string; then, it counts the number of the BSp strings to identify the end condition of the cross-hole (16 strings) which identifies the cross-hole is through. Then, the system reads the values of the vertex (1 and 2) of BSp and both values are ($\neq$0 mm), which indicates that the cross-hole is offset from the axis of the cylinder. Furthermore, the system reads the directions (Dx, Dy, and Dz) of BSp, and the values of two directions of these three are (>0 mm), which identifies that the cross-hole is inclined, and this means that the cross-hole is an inclined offset cross through-hole.

The software is also recognizing the inclined offset cross through-hole feature as shown in Figure 56 by applying the rules to recognize the cross-hole feature and showing a message box of the feature recognitions process.

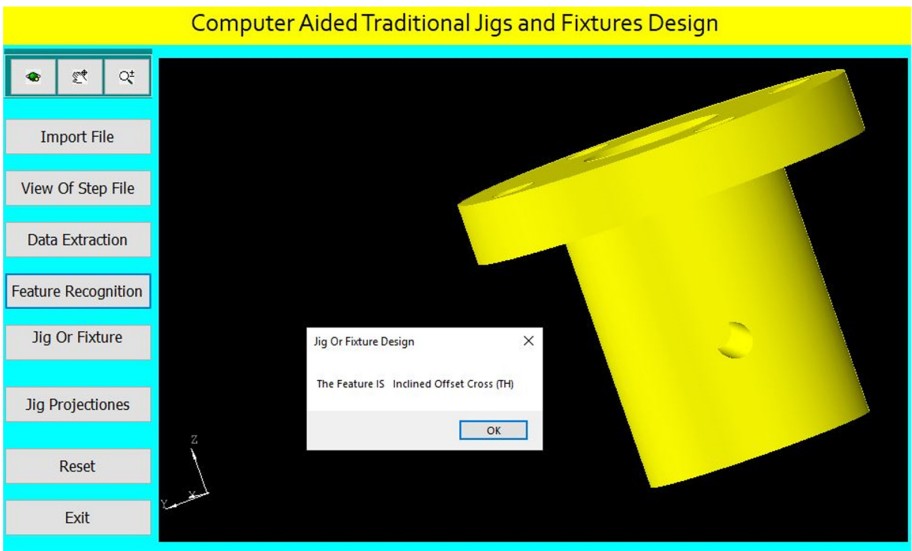

**Figure 56.** Automatic feature recognition of an inclined offset cross through-hole.

The program checks the inclined offset cross through-hole of (4 mm) diameter for identifying the suitable guide bush.

Design of the jig for this part is done parametrically on the software screen as shown in Figure 57.

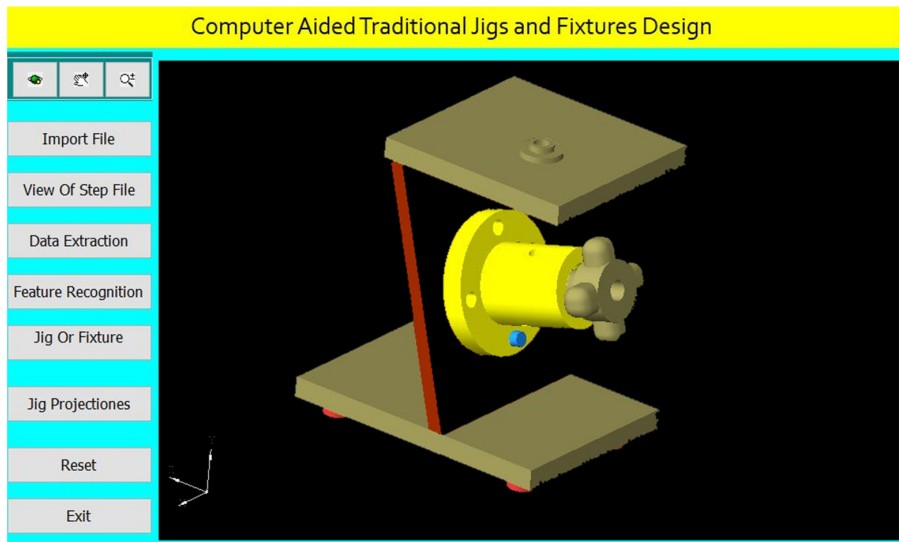

**Figure 57.** Computer-aided design of a jig for the cylindrical part.

The software shows the jig views in Figure 58. The button (Jig Projections) is used to draw the jigs views on the interface screen to show the details of the design of traditional jigs and fixtures.

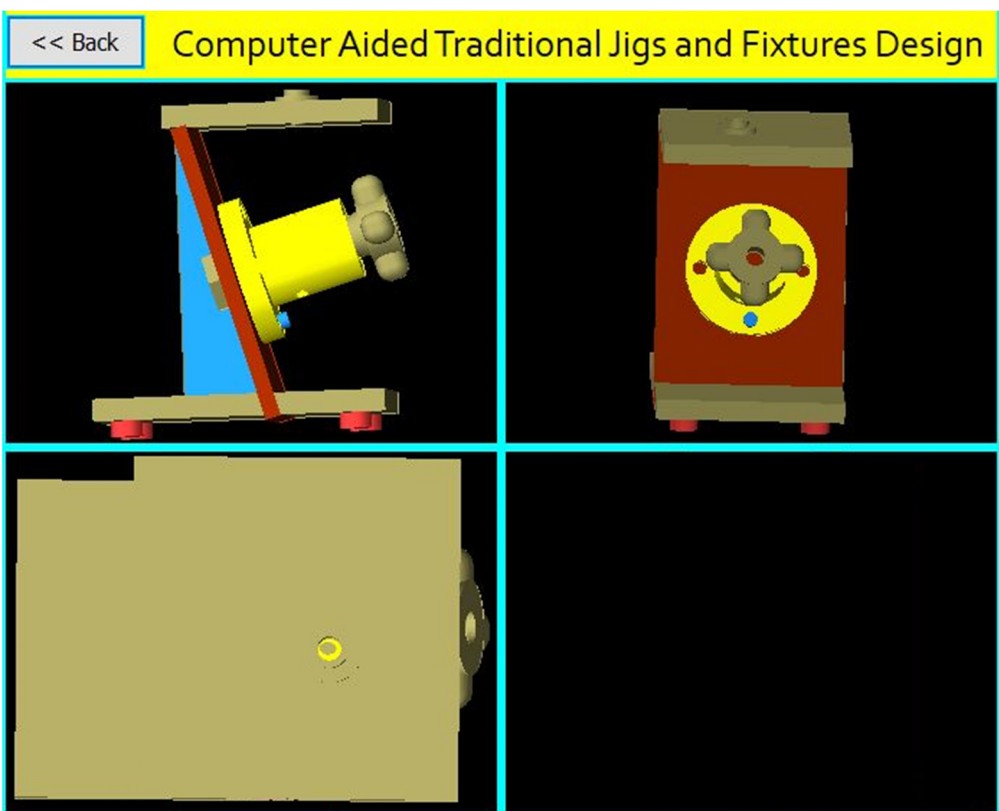

**Figure 58.** The jig views.

## 6. Design Verification

For verifying the jig design, the jig dimensions are compared to the design rules for the part which identify the following:

The jig height (H) = 2 Dmax (for the vertical jig).

The jig height (H) = 2Dmax/cos ($\theta$ rad) (for the inclined jig).

The jig width (W) = 1.5 Dmax.

The distance between the part and drill plate = 1.5 Dc.

The guide bush center is collinear with the cross-hole center.

## 7. Conclusions

Jigs and fixture design is a significant issue in the manufacturing process. Computer-aided traditional jigs and fixture design is a critical activity process for bridging CAD and CAM systems. This research presented the developed system for computer-aided traditional jigs and fixture design. This system extracts data from a STEP AP-203 file with full details of the part geometry, recognizes the different cases of the cross-holes in hollow cylinders, and provides an appropriate design of the traditional jigs and fixtures. The software has been built by connecting the Visual Basic programming language to the SolidWorks software, and this system has improved the design efficiency and reliability of the traditional jigs and fixtures and made the results of the design process more reasonable.

The developed system solves the problems of the manual design process by reducing the cost, time, and effort of the design process.

This system is of great importance for medium- and small-sized factories, and it is a good start for making an integrated system for all mechanical parts.

We recommend developing this system to design the traditional jigs and fixtures of the cylinders having intersected cross-holes.

**Author Contributions:** Conceptualization, H.M.H. and A.D.I.; methodology, A.D.I.; software, A.D.I.; validation, E.A.N., A.K. and S.A.A.; formal analysis, A.D.I.; investigation, H.M.H.; resources, H.M.H. and S.A.A.; writing—original draft preparation, A.D.I.; writing—review and editing, I.A.; visualization, E.A.N. and A.K.; supervision, and S.A.A.; project administration, E.A.N. and A.K.; funding acquisition, E.A.N. and A.K. All authors have read and agreed to the published version of the manuscript.

**Funding:** This research work is supported through the Researchers Supporting Project (RSP-2021/164), King Saud University, Riyadh, Saudi Arabia.

**Data Availability Statement:** Not applicable.

**Acknowledgments:** The authors extend their appreciation to King Saud University for funding this work through Researchers Supporting Project number (RSP-2021/164), King Saud University, Riyadh, Saudi Arabia.

**Conflicts of Interest:** The authors declare no conflict of interest.

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
