# Peer review of "Computer-Aided Design of Traditional Jigs and Fixtures"

_applsci, doi:10.3390/app12010003_

Round 1

Reviewer 1 Report

The pictures should have better clarity.

The article is a good read.

Author Response

Response to Reviewer (1)

Dear Respected Reviewer,

The authors would like to thank you very much for your precise review of our manuscript and for your constructive comments and suggestions to improve the presentation of our work. According to your comments, we have revised our manuscript as follows:

Note from authors: To make it easy following the changes, the comments of respected reviewers are highlighted in yellow.

Comment 1:

The pictures should have better clarity.

The Author response:

The unclear pictures have been sharpened.

Best regards

Reviewer 2 Report

The authors in this article describe a system developed for CAD for the design of jigs and fixtures. They show screenshots of the developed programming language.

I have had the opportunity to read the article in great detail and I personally consider that the article as presented lacks originality and innovation. it is not defined what the scientific contribution is. 
Furthermore, the literature review is too short for the length of the article. 

The way the results are presented is not considered appropriate for a research article for this journal. Also, the results should be presented in a more concise way with final data obtained and parameterisation equations for the computer-aided design.

It is advised to rewrite the article completely and analyse the results in a scientific way.

Author Response

Response to Reviewer (2)

Dear Respected Reviewer,

The authors would like to thank you very much for your precise review of our manuscript and for your constructive comments and suggestions to improve the presentation of our work. According to your comments, we have revised our manuscript as follows:

Note from authors: To make it easy following the changes, the comments of respected reviewers are highlighted in yellow.

Comment (1):

I have had the opportunity to read the article in great detail and I personally consider that the article as presented lacks originality and innovation. It is not defined what the scientific contribution is. 
Furthermore, the literature review is too short for the length of the article. 

Response to comment (1)

After reviewing the previous studies and research, it is clear that there are few pieces of research found that cover the computer-aided design of traditional jigs and fixture for cylindrical parts having cross-hole feature in various orientations, so, this is the reason of why the literature review is too short for the length of the article, furthermore, this indicates the originality and innovation of the current work.

Comment (2):

The way the results are presented is not considered appropriate for a research article for this journal. Also, the results should be presented in a more concise way with final data obtained and parameterization equations for the computer-aided design.

Response to comment (2)

The algorithm and technique of automatic feature recognition of cross-hole and computer-aided design of traditional jigs and fixtures is clearly explained through the flowcharts. For validating the methodology, eight case studies are explained.

Best Regards

Reviewer 3 Report

Please proof read article and watch out for the use of "of". It can be "deleted" in many instances. For example, "Increasing of global competition made all manufacturers …." in the "Introduction" (page 2) can be rendered as

"Increasing global competition made all manufacturers …."

or simply

"Increase in global competition made all manufacturers ....".

Author Response

Response to Reviewer (3)

Dear Respected Reviewer,

The authors would like to thank you very much for your precise review of our manuscript and for your constructive comments and suggestions to improve the presentation of our work. According to your comments, we have revised our manuscript as follows:

Note from authors: To make it easy following the changes, the comments of respected reviewers are highlighted in yellow.

Comment (1):

Please proof read article and watch out for the use of "of". It can be "deleted" in many instances. For example, "Increasing of global competition made all manufacturers …." in the "Introduction" (page 2) can be rendered as

"Increasing global competition made all manufacturers …."

or simply

"Increase in global competition made all manufacturers ....".

Response to Comment (1):

The whole article has been revised carefully and some sentences have been paraphrased as highlighted in yellow.

Best Regards

Reviewer 4 Report

The text needs significant revision. Language editing requires cohesion and coherence clarification and organization.  Several repetitions appear (cf. p.25 and 20) and there are many syntactic construction inconsistencies, dangling sentences and punctuation mistakes. In addition, some figures are repeated with a change only in the number sequence. For some drawings, it would be useful to specify the context of discussion. It would also be purposeful to describe with more precision how the model realized with SolidWorks and/or application protocols (AP) related to the representation of components and mechanical assemblages (AP3203) are connected to the actions of visual basic.

Author Response

Response to Reviewer (4)

Dear Respected Reviewer,

The authors would like to thank you very much for your precise review of our manuscript and for your constructive comments and suggestions to improve the presentation of our work. According to your comments, we have revised our manuscript as follows:

Note from authors: To make it easy following the changes, the comments of respected reviewers are highlighted in yellow.

Comment (1):

The text needs significant revision. Language editing requires cohesion and coherence clarification and organization.  Several repetitions appear (cf. p.25 and 20) and there are many syntactic construction inconsistencies, dangling sentences, and punctuation mistakes.

Response to Comment (1):

The whole article has been revised carefully, and language editing has been done. For cohesion and coherence clarification, many paragraphs have been rearranged.

Comment (2):

Some figures are repeated with a change only in the number sequence.

Response to Comment (2):

After a careful revision of the figures, it has been found that there is no repetition in figures. To make it short, some figures have been removed from the article.

 Comment (3):

For some drawings, it would be useful to specify the context of the discussion.

Response to Comment (3):

Some explanations have been added to the drawing’s discussion.

Comment (4):

It would also be purposeful to describe with more precision how the model realized with SolidWorks and/or application protocols (AP) related to the representation of components and mechanical assemblages (AP203) are connected to the actions of visual basic.

Response to Comment (4):

SolidWorks software is used to just prepare the part model and save it as a STEP AP-203 file. Other CAD software such as AutoCAD or CATIA can also be used for preparing the part model and saving it as a STEP AP-203 file. Our developed system reads the STEP AP-203 file and extracts the geometrical data, recognizes the cross-hole feature, and provides the appropriate design of traditional jigs and fixtures according to the design rules explained clearly through the flowcharts.

 Best regards

Reviewer 5 Report

This is a paper discussing a methodology aimed towards designing jigs and fixtures using Computer-Aided-Design (CAD). Overall, the paper is too descriptive, contains multiple typos and requires sentence restructuring at several places for ease to the reader. Some examples right at the beginning of the manuscript:

1. The traditional designing method of jigs and fixtures became not suitable for ... (Conventional design of jigs and fixtures has become unsuitable given the ... )
2. Computer Aided Design (CAD) of jigs and fixtures is an effective way for this case. ( ... in this direction).
3. The current paper focuses on a computer-aided design of the traditional jigs and fixtures, and developed a system contains a tailor-made software, created using the Visual Basic programming language and installed on it the viewer screen to show the part (sentence restructuring needed: ... developed a system containing a tailor-made software ... )
4. Jigs and Fixtures are devices which widely used in manufacture, (which are widely used in manufacturing)
5. The efficient and reliable design and manufacture of jigs and fixtures became more required with the application of CAM technology. (CAM has not been defined here, sentence restructuring needed)

... and multiple such others throughout the manuscript. It makes it quite difficult to even read the manuscript. Editors, please take a note.

In Section 2, Literature review, the authors do not compare what value their technique adds in comparison to the previous methodologies in literature. Strongly suggest making a table comparing the pros and cons of their proposed technique to existing methods. What value does your proposed technique add over those existing in literature ?

Please reduce unnecessary figures, such as, Figure 7, Figure 8 and several such figures are not needed for the benefit of the reader. Clearly state your technique and how it differs from the existing techniques in literature. It is not important to discuss how the software tool operates, but rather what merits your technique provides.

In short, this manuscript requires significant corrections and work. It requires significant corrections before it can be considered for publication.

Author Response

Response to Reviewer (5)

Dear Respected Reviewer,

The authors would like to thank you very much for your precise review of our manuscript and for your constructive comments and suggestions to improve the presentation of our work. According to your comments, we have revised our manuscript as follows:

Note from authors: To make it easy following the changes, the comments of respected reviewers are highlighted in yellow.

Comment (1):

Overall, the paper is too descriptive, contains multiple typos, and requires sentence restructuring at several places for ease to the reader. Some examples are right at the beginning of the manuscript.

Response to Comment (1):

The whole article has been revised carefully and the requested corrections have been made and some sentences have been paraphrased as highlighted in yellow.

Comment (2):

In Section 2, Literature review, the authors do not compare what value their technique adds in comparison to the previous methodologies in literature. Strongly suggest making a table comparing the pros and cons of their proposed technique to existing methods. What value does your proposed technique add over those existing in literature?

Response to Comment (2):

The literature review has been put in the form of a table showing the methodology and remarks of each study and the authors have added a comment explaining what value the proposed technique adds over those existing in the literature.

Comment (3):

Please reduce unnecessary figures, such as Figure 7, Figure 8, and several such figures are not needed for the benefit of the reader.

Response to Comment (3):

For the benefit of the reader and to make it shorter, some figures have been to removed from the article.

Comment (4):

Clearly state your technique and how it differs from the existing techniques in the literature. It is not important to discuss how the software tool operates, but rather what merits your technique provides.

Response to Comment (4):

The literature review has been edited and a comment explaining what value the proposed technique adds over those existing in the literature.

Best regards

Round 2

Reviewer 5 Report

I would recommend a careful reading to the authors to correct further typos. Everything else looks good.